# QRazor: Reliable and Effortless 4-bit LLM Quantization by Significant Data Razoring

## Abstract

Large-scale language models (LLMs) have demonstrated outstanding performance in language processing tasks, yet their deployment is often hindered by high memory demands and computational complexity. Although low-bit quantization techniques, such as 4-bit quantization, present a potential solution, they frequently lead to significant accuracy degradation or require substantial effort for such aggressive quantization approaches. To overcome these challenges, we introduce QRazor, a reliable and effortless quantization scheme designed to enable 4-bit quantization for weights, activations, and KV cache in transformer-based LLMs. The scheme involves two main stages: quantization and compression. During the quantization stage, weights, activations, and KV cache values are quantized with wider 8 or 16-bit integers as a basis to achieve nearly identical accuracy to the original full-precision LLM models, using the absolute max scaling. Subsequently, all data are compressed to 4-bit using our proposed significant data razoring (SDR) technique, which retains only the four most salient bits while discarding the others. Furthermore, we present an integer-based arithmetic unit dedicated to QRazor, enabling direct low-precision arithmetic operations without decompressing the SDR data. Despite the reduced quantization effort, QRazor achieves LLM accuracies better or comparable to state-of-the-art 4-bit methods. By also validating the hardware efficiency, our decompression-free arithmetic unit achieves 61.2% and 57.8% reduction in area and power consumption, respectively.

## 1 Introduction

Large language models (LLMs) have garnered considerable attention for their exceptional performance across various domains, including natural language processing and automated text generation. However, the extensive number of parameters in LLMs poses challenges for efficient processing. In particular, the substantial memory requirements and high computational demands can hinder their deployment in resource-constrained computing environments.

Quantization (Dettmers et al., 2022),(Yao et al., 2022) is widely recognized as one of the most effective techniques for addressing this challenge. By aggressively reducing the precision of LLM parameters from 16-bit floating-point (FP16) to low-bit integers such as 4-bit, both memory usage and computational complexity can be significantly decreased during inference. Quantization methods are broadly divided into Post-Training Quantization (PTQ)(Banner et al., 2019) and Quantization-Aware Training (QAT)(Liu et al., 2023). This work focuses on PTQ, due to the substantial computational cost of retraining LLMs required by QAT. For low-bit integer PTQ of LLM, the key challenge is a significant accuracy drop. According to previous research (Xiao et al., 2024; Lin et al., 2024a; Yuan et al., 2023; Ashkboos et al., 2024), quantizing activations to low-bit integers is more challenging than quantizing weights due to the wider dynamic range and the presence of outliers in activations. As a result, these studies have proposed various methods to identify and mitigate activation outliers during the calibration phase of PTQ.

For instance, QLLM (Liu et al., 2024a) introduces a channel reassembly technique to minimize quantization errors in activations. However, these techniques still face significant accuracy degradation when both weights and activations are quantized to 4-bit integers. A recent study, Quarot (Ashkboos et al., 2024), successfully quantized both weights and activations to 4-bit integers while maintaining reasonable accuracy in LLMs. Notably, Quarot also demonstrated the feasibility of quantiz-

ing KV caches down to 3-bit, significantly reducing memory usage. To achieve this, Quarot utilizes orthogonal Hadamard-based rotations (Steinerberger, 2024) to suppress outliers in activations and KV caches, effectively mitigating the impact of outliers in activations. However, Hadamard-based rotation does not always guarantee effective outlier suppression despite its computation overhead. Furthermore, since the efficacy of these rotations is highly dependent on the underlying data distribution, applying Quarot to models with multiple data distributions, such as in mixture-of-expert models, requires significant tuning efforts.

This paper introduces QRazor, a PTQ method to deliver reliable 4-bit LLMs. The QRazor employs two key insights: 1) Utilizing 8-bit integers for weights and KV cache values, combined with 16-bit integers for activations, effectively captures data distributions, including outliers, and serves as the base precision scenario in this work. 2) By capturing a few salient bits from the base precision scenario, we can preserve essential characteristics of both outliers and non-outliers at reduced bit precision.

Building on these insights, QRazor operates in two stages: quantization and compression. In the quantization stage, we first quantize weights, activations, and KV caches according to the base precision scenario described earlier. Our experiments demonstrate that in this PTQ setup, the accuracies of various LLMs remain nearly identical to those of their original counterparts. Next, the parameters are compressed to our target precision using our significant data razoring (SDR) technique. Our SDR technique captures a few salient bits and discards the other bits, implemented through bitwise operations, truncation, and round-to-nearest. Such a scheme does not manipulate data distributions and is easily adaptable to various models.

This work considers the following two scenarios based on our QRazor scheme: W4A4 (4-bit weights, 4-bit activations, FP16 KV cache) and W4A4KV4 (4-bit weights, 4-bit activations, 4-bit KV cache). The accuracies of these configurations are compared against state-of-the-art (SOTA) 4-bit LLMs. For the LLaMA-1-7B and LLaMA-13B models, QRazor demonstrates superior performance, achieving more than a 10% improvement in accuracy compared to QLLM Liu et al. (2024a). When compared to Quarot, QRazor achieves significantly better results when evaluated against the rounding-to-nearest (RTN) baseline, and nearly matching results for configurations involving GPTQ for the LLaMA-2-7B and LLaMA-13B models. For other models, such as Mistral-7B and GEMMA2-2B, QRazor achieves strong accuracy results. These findings demonstrate QRazor's ability to deliver consistent and reliable performance across diverse LLM models.

Additionally, we develop an integer-based arithmetic unit specifically optimized for QRazor, enabling decompression-free computations and eliminating the hardware overhead associated with group-level dequantization. The multiplication of 4-bit compressed data is performed using a 4-bit integer multiplier, significantly improving computational efficiency. Hardware simulations show that our design, implemented for scenarios such as W4A4 or W4A4KV4, achieves 57.8% power and 61.2% area savings compared to arithmetic operations performed after decompression.

## 2 RELATED WORK

**LLM quantization.** Due to the wide quantization range required for activations, most previous PTQ schemes have focused on weight-only quantization, where weights are reduced to extremely low bits while activations remain in FP16 format (Frantar et al., 2023; Dettmers et al., 2023; Lee et al., 2024; Kim et al., 2024; Chee et al., 2024; Cheng et al., 2024). These schemes are optimized to minimize memory consumption, yet computations are still performed in high precision. To reduce the precision of both weights and activations, various studies have been conducted on mitigating activation outliers by balancing the data range (Wei et al., 2023; Zhang et al., 2023; Li et al., 2023b; Behdin et al., 2023; Shao et al., 2024; Li et al., 2023a). Furthermore, more fine-grained quantization strategies, such as channel-wise quantization (Wang et al., 2024) and group-wise quantization (Dai et al., 2021; Yao et al., 2022), have been explored to achieve reliable results in LLM services.

Recently, several works have succeeded in quantizing both weights and activations to 4-bit integers (Guo et al., 2023; Ashkboos et al., 2024). Atom (Zhao et al., 2024) and QUIK (Ashkboos et al., 2023) achieve this by quantizing most of the data to 4 bits while retaining a small portion of data in 8-bit form. QLLM (Liu et al., 2024a) introduces a channel reassembly technique, and QuaRot (Ashkboos et al., 2024) utilizes Hadamard-based rotations to suppress outlier issues and en-

able effective 4-bit inference. Meanwhile, KV cache quantization is another crucial research topic, especially for LLMs (Sheng et al., 2023; Hooper et al., 2024), as the cache size becomes a major memory bottleneck when operating with large batches or generating long contexts. Notably, QuaRot (Ashkboos et al., 2024) demonstrates 4-bit quantization of all the data, including the KV cache.

**Outlier mitigation.** Recent studies have shown that smoothing activation outliers by transitioning magnitudes between activations and weights can effectively reduce the wide quantization range (Xiao et al., 2024). Several studies have mitigated outliers using reorder-based quantization by clustering the activation channels (Yuan et al., 2023; Zhao et al., 2024). These studies have reduced addressing complexity by fusing the reorder operation into the previous layer normalization operation. Other approaches, which apply transformations to the matrices to reduce outlier magnitudes, have also succeeded in reducing the entire weight and activation data to low-bit integers (Liu et al., 2024a; Ashkboos et al., 2024).

There are several encoding methods equipped with dedicated hardware support that effectively manage outliers to be processed with the same precision as non-outlier data (Ho et al., 2020). OliVe (Guo et al., 2023) proposes the outlier-victim pair mechanism, which provides extra representation space for outliers, thereby maintaining the same precision for all data. O2A (Ho et al., 2020) encodes the entire data into low precision by leveraging additional flag bits. These flag bits, produced per group, indicate the magnitude of the outlier values. The uniformly compressed bits are then dequantized based on the flag bits for computation.

## 3 PRELIMINARIES: ABSOLUTE MAX SCALING (ABS)

The absolute max scaling (Latotzke et al., 2022) ensures that input data fits within a target bit-width by utilizing the absolute maximum value of the data. The quantization process begins by identifying the absolute maximum value $|X_{\max}|$ in the tensor, which is then used to calculate the scale factor. For example, in the case of 8-bit integer quantization, the formula to quantize a tensor $X$ under absolute max scaling and its dequantization formula are given by:

$$X_q = \text{round}(\frac{127}{|X_{\max}|} \cdot X), \qquad \hat{X} = \frac{|X_{\max}|}{127} \cdot X_q$$

where $X_q$ and $\hat{X}$ are the quantized and dequantized tensors, respectively. This scaling guarantees that all values in the tensor are normalized within the 8-bit range, maximizing the precision within the allowed bit-width. The absolute max scaling is applied to establish the base precision scenario outlined in Section 1, as it is known to be less sensitive to outliers compared to the min-max scaling method (Patro & Sahu, 2015; de Amorim et al., 2023).

## 4 THE QRAZOR SCHEME

Our QRazor scheme consists of two stages: quantization and compression. In the quantization stage, we first quantize FP16 parameters to high-bit integers to accurately represent LLM parameters without any loss of accuracy, which we refer to as the base precision scenario. It is important to note that, at this stage, only the static quantization method (Xiao et al., 2024) has been employed. Following this, the integers in the base precision format are compressed to lower-bit data using our SDR technique.

Figure 1 illustrates the concept of our QRazor scheme, specifically detailing the process of quantizing activations to 4-bit. Most 4-bit PTQ schemes divide the tensor data into multiple groups. After grouping, the data in each group (initially in FP16 precision) are directly quantized to 4-bit using a single scale factor with FP16 precision. In contrast, our QRazor scheme quantizes the entire tensor data to 16-bit integers during the quantization stage, with a single scale factor shared across the entire tensor. Afterward, the quantized tensor is divided into multiple groups, and the 16-bit integer data in each group are compressed to 4-bit using our SDR technique.

It is important to emphasize that in the quantization stage of our QRazor scheme, activations and KV-caches utilize per-tensor and per-head scaling, respectively, while weights employ per-channel scaling. All parameters use static scaling at this stage. In contrast, Quarot, the SOTA method for

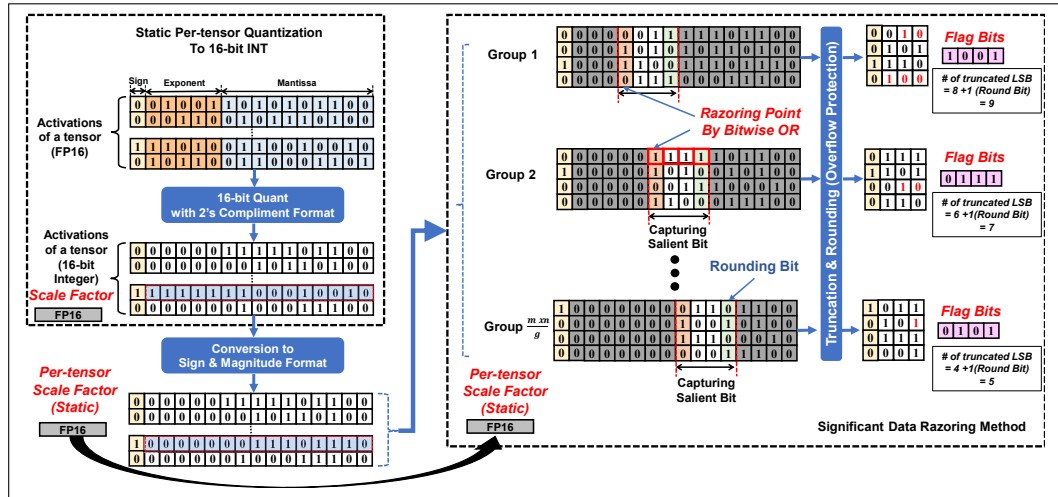

Figure 1: The overall concept of (a) typical integer-based quantization and (b) our QRazor scheme.

quantizing all LLM parameters to 4-bit, similarly uses per-channel scaling for weights but adopts per-token scaling for activations and per-group scaling for KV caches, with a group size of 128, to enhance LLM accuracy in low-bit precision scenarios (Ashkboos et al., 2024). Notably, Quarot also relies solely on static scaling methods.

Unlike weights, which are quantized offline, activations and KV caches must be quantized online using static scaling parameters. In this context, the scaling granularity of activations and KV caches plays a critical role in determining the computational complexity of LLM inference. Our coarse granularity scaling approach—per-tensor for activations and per-head for KV caches—reduces the computational complexity compared to methods employing fine granularity scaling, such as per-token for activations and per-group for KV caches in Quarot (Xiao et al., 2024). This coarse granularity effectively offsets the relative overhead introduced by compression, as discussed in the following subsections. Moreover, despite the inherent limitations of coarse granularity, our QRazor scheme achieves accuracies that are either superior or comparable to Quarot, as discussed in Section 5.1.

### 4.1 QUANTIZATION STAGE: W8A16 OR W8A16KV8 INTEGER QUANTIZATION

In the quantization stage, LLM parameters are converted from FP16 to two base precision scenarios: W8A16 (8-bit weights, 16-bit activations) and W8A16KV8 (8-bit weights, 16-bit activations, 8-bit KV caches). These base precisions serve as primitive data types in the subsequent compression stage. Specifically, W8A16 forms the basis for W4A4, while W8A16KV8 serves as the basis for W4A4KV4, where all parameters, including KV caches, are quantized. We assume that the base precisions utilize a 2's complement format to represent negative numbers.

As mentioned in Section 2, outliers present challenges for low-bit LLM quantization. Activations typically exhibit a more comprehensive dynamic range than weights or KV caches (Lin et al., 2024b), resulting in more extreme outliers observed within. Hence, quantizing activations effectively requires using a higher-precision format, such as 16-bit integers, to adequately capture the full range of values without significant loss of information. In contrast, we can successfully express weights and KV caches using only 8-bit integers without hardly affecting LLM accuracies.

We evaluated LLM accuracies for three scenarios: W8A8 (8-bit weights and 8-bit activations), W8A16, and W8A16KV8. These results support the hypothesis mentioned above well. Our base precision scenarios, W8A16 and W8A16KV8, achieve nearly identical accuracies to FP16-based models across various LLaMA tasks. In contrast, the quantization using W8A8 leads to significant accuracy drops, clearly demonstrating that 16-bit integers effectively characterize activation outliers, while 8-bit integers fail to do so (See Appendix A.1 for details).

**Algorithm 1** QRazor Compression of W/A/KV with Exception Handling by Group Size g.

$func$ QRazor($X_{i,j}, w, t, g$)

  **Input**:
      $X_{i,j} \in \mathbb{R}$ — Weight, Activation, KV cache matrix,
      $b_w$ — Quantization bit width
      $b_t$ — Total bits to remove for compression
      $g$ — Element group size
  **Initialize**:
      $b_k = b_w - b_t$           // compressed bit width
  **Algorithm Steps**:
      $E_{i,j} = \text{Quantize}(X_{i,j}, b_w)$
      For each element $e_{i,j}$ in $E_{i,j}$
      $s_{i,j} \leftarrow$ sign bit of $e_{i,j}$       // extract sign bit
      **for** $i = 1, 2, ..., m$ **do**
          **for** $j = 1, 2, ..., n$ **do**
             **if** $e_{i,j} < 0$ **do**
                $e_{si_{i,j}} = \text{remove\_sign\_bit}(e_{i,j})$
                $e_{ci_{i,j}} = \text{concat}(s_{i,j}, \text{2's\_complement}(e_{si_{i,j}}))$   // calculate 2's complement value
            **else do**
                $e_{ci_{i,j}} = e_{i,j}$
      For each element $e_{ci_{i,j}}$ in $E_{ci_{i,j}}$
      $G_p = \text{group\_by\_size\_g}(e_{ci_{i,j}}, g)$
      **for** $p = 1, 2, ..., \dfrac{m \cdot n}{g}$ **do**       // ($\dfrac{m \cdot n}{g} \in \mathbb{Z}^+$)
          $b_m = \text{detect\_redundant\_MSBs}(G_p, b_t, b_w)$   // detect redundant MSBs bit width to truncate
          $b_l = b_t - b_m$       // calculate LSBs bit width to truncate
          $F_{flagbit} = b_l$
          $G_q = \text{truncate\_MSBs\_and\_LSBs}(e_{ci_{i,j}}, b_m, b_l)$   // truncate $m$ MSBs after the sign bit & truncate $l$ LSBs
          **if** $G_q = 2^k - 1$ **do**
             $G_r \leftarrow G_q^{\text{floor}}$       // truncate LSBs without carry to salient bits
          **else do**
             **if** MSB of truncated LSBs $= 0$ **do**
                $G_r \leftarrow G_q^{\text{floor}}$
             **else do**
                $G_r \leftarrow G_q^{\text{ceil}}$     // $r \in \{0, 1, 2, ..., \dfrac{m \cdot n}{g}\}$, truncate LSBs with carry to salient bits
      $C = \{G_1, G_2, ..., G_{\frac{m \cdot n}{g}}\}$
      **return** $C, F_{flagbit}$

## 4.2 COMPRESSION STAGE: SIGNIFICANT DATA RAZORING

Algorithm 1 shows the pseudo-code of our QRazor scheme, where the detailed operation of the compression stage is explained. At the beginning of the compression, the base precision tensors with 2's complement format are converted to the tensor with sign and magnitude format. This can be easily obtained by processing 2's complement for negative values while preserving their sign bits.

Following this, our SDR technique identifies the "razoring point" for each group, which corresponds to the bit position of the leading one. The razoring point is determined by detecting the bit position of the leading one from the bitwise OR result of all data within the group (See Appendix A.3 for details). Once the razoring point is identified, a defined number of adjacent bits starting from this point, known as the salient bits, are captured. The bit width of these salient bits matches the target precision. For instance, to achieve 4-bit activations, the width of the salient bits must be four, as illustrated in Figure 1.

After capturing salient bits, we retain only the sign bit and the selected salient bits, truncating all the other bits. Then, by rounding the last bit of the remaining bits, we obtain the final compressed values. In such a scheme, during the rounding process of the least significant bits (LSBs), there is a potential risk of overflow. For example, rounding up "$01111_2$," where the most significant bit (MSB) represents the sign, could alter the sign bit. To prevent this, we avoid rounding the LSBs of elements where all salient bits are '1'. Instead, we apply flooring to the LSBs of these elements while continuing to round the LSBs of other elements within the group. The detailed procedure for handling the above exception is outlined in Algorithm 1.

Ultimately, our SDR technique eliminates the '0's in higher bit positions above the razoring point, which can be reconstructed using flag bits that indicate the razoring point for each group — the flag bits represent the number of truncated LSBs in the group and thus inform the razoring point when the base precision is known. We validate the efficacy of our SDR technique across various scenarios, demonstrating that it effectively compresses all data into a 4-bit format while delivering reliable accuracies across multiple LLM tasks, as further discussed in Section 5.1.

The size of the compression group can vary from 16 to 128. The reason we can accommodate relatively small group sizes during compression is that our SDR technique does not involve any computational quantization but simply removes zeros from the MSB portion of the integer-based data, along with a straightforward rounding process. As a result, unlike other group-based quantization approaches, there is no need for a scaling process per group using higher precision. This enables direct computation with low-bit operands, supported by our dedicated arithmetic design. The design facilitates both memory- and compute-efficient low-bit matrix multiplication, with hardware details provided in Section 4.3. It is important to note that the FP16 scaling process in QRazor is applied only per tensor for activations, per channel for weights, and per head for the KV cache.

Due to the aggressive 4-bit compression, one might be concerned that the number of zeroed elements significantly increases, degrading the reliability of our QRazor. Figure 2(a), (b) demonstrates that the first concern is effectively addressed in our QRazor. It illustrates the portion of leading '1' positions before 4-bit compression (immediately after converting to the sign-and-magnitude format). The leading '1' positions for activations are predominantly located between the 8th and 12th bit orders. For instance, if the leading '1' position in a group is the 13th bit, parameters with an MSB below the 9th bit will be rounded and zeroed after compression, as only 4 bits are retained. Fortunately, the number of groups where the leading '1' position exceeds the 12th-bit order is minimal (only 9%), indicating that outliers are infrequent and mitigating concerns about significant parameters being truncated after 4-bit compression.

In Figure 2(c) we also analyzed the proportion of zeroed elements before and after 4-bit compression. For Query, Key, and Value, the increase in zeroed elements is not so considerable. However, for activations and weights, there is a notable increment. This is expected, as significant activations and weights often have small absolute values close to zero. Such truncation of small values does not lead to substantial errors. Combined with the low occurrence of outliers, as mentioned above, this explains why QRazor consistently delivers reliable performance across various LLMs, as discussed in Section 5.1.

It is important to note that in our compression, the truncation level dynamically varies for each group during runtime. Due to this characteristic, one might draw a comparison between our compression and dynamic max-scaled quantization (DMQ). However, our compression technique fundamentally differs from DMQ in the following aspects:

**1. Eliminating Absolute Max Computation:** Instead of determining the absolute maximum value within a group, QRazor detects only the leading '1'. While the absolute maximum inherently contains the leading '1', multiple parameters may share the same leading '1' position. In DMQ, identifying the absolute maximum value which is presented in floating point data format manner, is essential for computing the scaling factor. In contrast, our compression bypasses this step entirely, significantly reducing computational complexity.

**2. Lightweight Compression and Decompression:** Per-group DMQ typically involves group-level quantization and dequantization operations, which rely on arithmetic computations such as multiplication and, in some cases, division. In contrast, our compression and decompression rely on bit-level truncation and shifting, which are inherently simpler and far less resource-intensive.

To conclude, while DMQ and our compression share the goal of dynamically reducing precision, their underlying methodologies are fundamentally different. As outlined above, QRazor's compression incurs substantially lower computational overhead than DMQ. Moreover, QRazor's simpler operations require fewer hardware resources, enabling more efficient implementation in dedicated arithmetic units, as further discussed in the following section.

### 4.3 DECOMPRESSION-FREE ARITHMETIC OPERATION

We introduce an arithmetic unit designed to execute multiply-and-accumulate (MAC) operations required for matrix multiplication while preserving the compression of our QRazor, a method we refer to as decompression-free arithmetic. Typically, data compressed by QRazor would need to be decompressed to their base precisions before performing MAC operations. This additional decompression process can introduce significant area overhead and degrade throughput, ultimately reducing performance when deployed on hardware devices. To address this challenge, we propose a decompression-free integer-based arithmetic unit, which enhances throughput by directly comput-

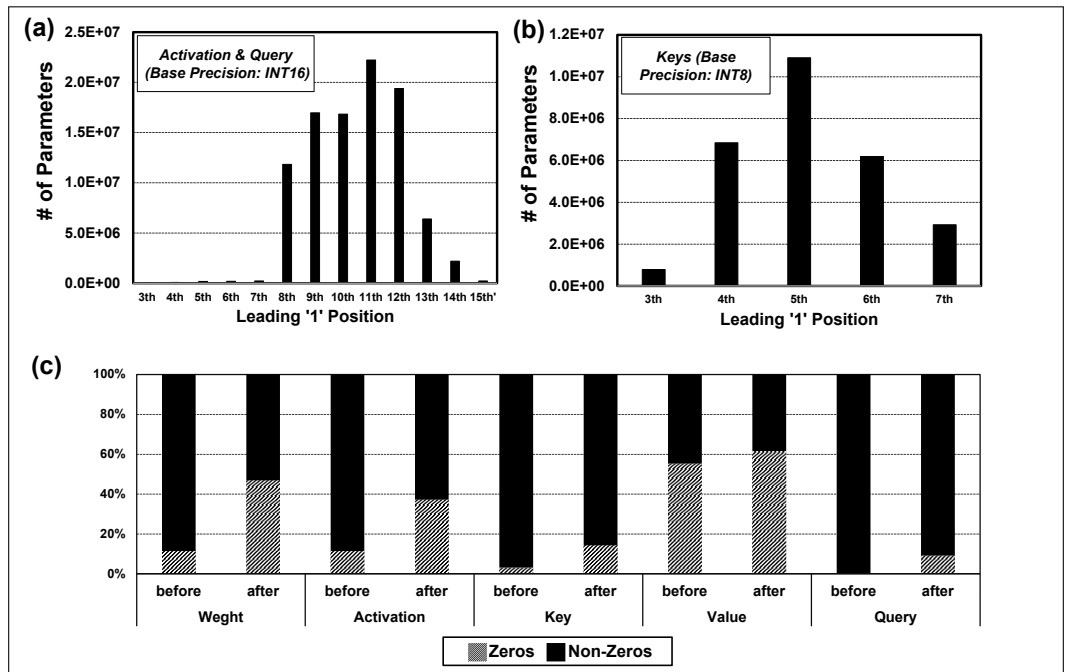

Figure 2: Leading '1' Position Before the Compression for (a) Activation and Query, and (b) Key. (C) Portion of Zeroed Elements between "Before 4-bit compression" and "After 4-bit compression"

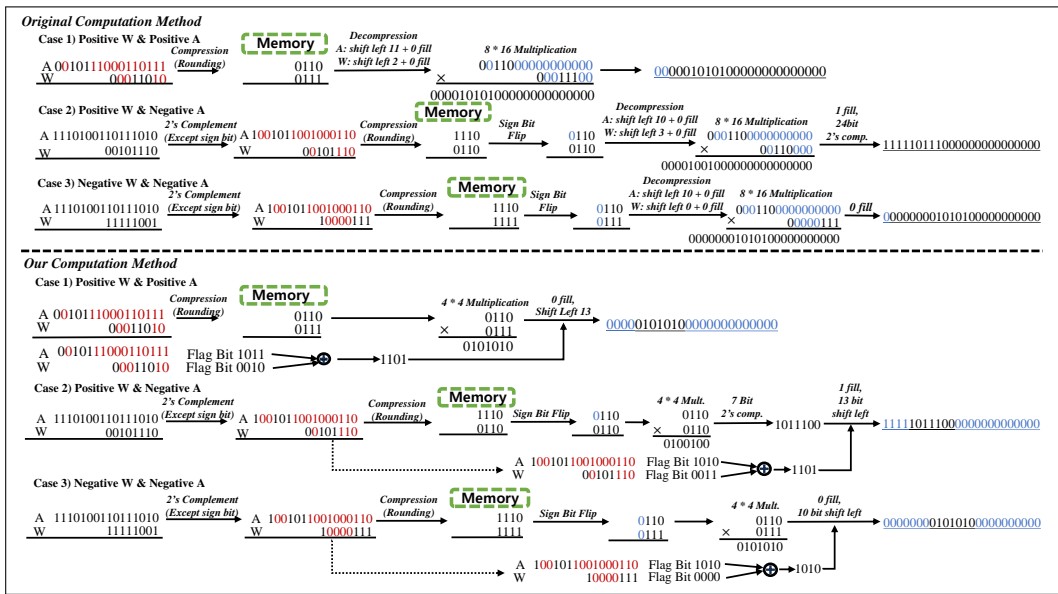

Figure 3: The concept of decompression-free arithmetic compared with the conventional method.

ing with the compressed low-precision data. This is achieved and evaluated by implementing the dedicated processing unit using a 4×4 multiplier and a single 16-bit barrel shifter.

The top part of Figure 3 illustrates the concept of conventional arithmetic operations for QRazor, which involve decompression of operands before computation. In this approach, weight and activation values are decompressed separately to their base precision using logical shift operations, followed by multiplication with a high-precision of 16×8 multiplier. In contrast, the proposed decompression-free arithmetic operation is shown in the bottom part of Figure 3. Here, the operands are directly multiplied using the 4×4 multiplier. The multiplied results are then manipulated by the

Table 1: Zero-shot accuracy of W4A4 LLaMA models on five common sense tasks.

| Model | #Bits | Eff. Bits | Method | PIQA | ARC-e | ARC-c | HellaSwag | Winogrande | Avg |
|---|---|---|---|---|---|---|---|---|---|
| | | | | | | | Zero-shot Accuracy ↑ | | |
| LLaMA-1-7B | FP16 | 16 | - | 77.67 | 76.73 | 40.13 | 73.08 | 68.78 | 67.28 |
| | W4A4 | 4.03/4 | OS+ | 62.73 | 39.98 | 30.29 | 44.39 | 52.96 | 46.07 |
| | W4A4 | 4 | OmniQuant | 66.15 | 45.20 | 31.14 | 56.44 | 53.43 | 50.47 |
| | W4A4 | 4 | QLLM | 68.77 | 53.16 | 31.14 | 57.43 | 56.67 | 51.84 |
| | W4A4 $g16$ | 4.25 | QRazor | 76.61 | 73.33 | 37.79 | 72.26 | 66.93 | 65.38 |
| | W4A4 $g32$ | 4.125 | | 75.24 | 69.47 | 36.12 | 70.98 | 66.69 | 63.70 |
| LLaMA-1-13B | FP16 | 16 | - | 79.16 | 79.65 | 43.48 | 79.07 | 72.77 | 70.83 |
| | W4A4 | 4.03/4 | OS+ | 63.00 | 40.32 | 30.38 | 53.61 | 51.54 | 47.77 |
| | W4A4 | 4 | OmniQuant | 69.69 | 47.39 | 33.10 | 58.96 | 55.80 | 52.99 |
| | W4A4 | 4 | QLLM | 71.38 | 47.60 | 34.30 | 63.70 | 59.43 | 55.28 |
| | W4A4 $g16$ | 4.25 | QRazor | 77.53 | 77.72 | 40.13 | 76.12 | 69.53 | 68.21 |
| | W4A4 $g32$ | 4.125 | | 76.17 | 74.56 | 37.12 | 74.45 | 67.64 | 65.99 |
| LLaMA-2-7B | FP16 | 16 | - | 79.13 | 74.39 | 45.97 | 76.21 | 69.30 | 69.00 |
| | W4A4 | 4.03/4 | OS+ | 63.11 | 39.10 | 28.84 | 47.31 | 51.30 | 45.93 |
| | W4A4 | 4 | OmniQuant | 65.94 | 43.94 | 30.80 | 53.53 | 55.09 | 49.86 |
| | W4A4 | 4 | QLLM | 67.68 | 44.40 | 30.89 | 58.45 | 56.59 | 51.60 |
| | W4A4KV4 | 4/4/4.125 | QuaRot(RTN) | 72.09 | 58.88 | 35.24 | 65.40 | 60.69 | 58.26 |
| | W4A4KV4 | | QuaRot(GPTQ) | 76.77 | 69.87 | 40.87 | 72.16 | 63.77 | 65.64 |
| | W4A4 $g16$ | 4.25 | QRazor | 75.84 | 72.63 | 42.47 | 72.96 | 65.67 | 65.91 |
| | W4A4 $g32$ | 4.125 | | 73.67 | 70.70 | 39.46 | 71.46 | 64.09 | 63.88 |
| | W4A4KV4 $g16$ | 4.25 | | 73.39 | 70.88 | 39.80 | 70.15 | 64.01 | 63.65 |
| | W4A4KV4 $g32$ | 4.125 | | 73.23 | 67.54 | 37.46 | 67.16 | 60.46 | 61.17 |
| LLaMA-2-13B | FP16 | 16 | - | 80.23 | 77.82 | 48.76 | 79.39 | 72.30 | 71.70 |
| | W4A4 | 4.03/4 | OS+ | 64.47 | 41.46 | 32.17 | 59.30 | 51.38 | 49.76 |
| | W4A4 | 4 | OmniQuant | 69.80 | 47.22 | 33.79 | 59.34 | 55.49 | 53.13 |
| | W4A4 | 4 | QLLM | 70.46 | 48.48 | 34.39 | 62.80 | 55.41 | 54.31 |
| | W4A4KV4 | 4/4/4.125 | QuaRot(RTN) | 77.37 | 70.83 | 43.69 | 73.11 | 67.32 | 67.16 |
| | W4A4KV4 | | QuaRot(GPTQ) | 78.89 | 72.98 | 46.59 | 76.37 | 70.24 | 69.79 |
| | W4A4 $g16$ | 4.25 | QRazor | 77.48 | 75.09 | 43.47 | 76.83 | 70.09 | 68.59 |
| | W4A4 $g32$ | 4.125 | | 76.82 | 73.86 | 41.80 | 76.30 | 68.59 | 67.47 |
| | W4A4KV4 $g16$ | 4.25 | | 77.80 | 75.44 | 44.15 | 76.06 | 67.72 | 68.23 |
| | W4A4KV4 $g32$ | 4.125 | | 77.09 | 74.91 | 44.28 | 75.06 | 66.77 | 67.62 |

\* Note: In the case of quantization granularity of channel and tensor, scale factor overhead is ignored.

flag bits using a single 16-bit barrel shifter. This approach eliminates the need for separate decompression steps, thereby enhancing computational efficiency and throughput. The hardware efficiency is further analyzed in Section 5.3.

## 5 EXPERIMENTS

We evaluate our proposed QRazor method on the LLaMA(Touvron et al., 2023a) and LLaMA2(Touvron et al., 2023b) models. Static quantization utilizing the absolute max scaling is implemented for all weights, activations, and KV cache values, with granularity configured as per-channel for weights, per-tensor for activations, and per-head for KV cache values. The SDR technique is applied for 4-bit compression using various group sizes. Weight compression is performed offline, while activations and KV caches are compressed online. Calibration and validation of zero-shot accuracy and perplexity were performed by randomly selecting 128 samples from Wikitext2 for all tasks, with perplexity evaluated on LAMBADA-OpenAI and accuracy on PIQA, ARC, Hellaswag, and Winogrande using lm-evaluation-harness (Gao et al., 2023). Additionally, an ablation study was conducted to assess the impact of group size on quantization accuracy, focusing on the compressed bits for weights and activations. A sequence length of 2048 and a batch size of 1 were used for all tasks during evaluation, and the models provided by meta-llama were employed for the tests.

### 5.1 LLM RESULTS

**LLAMA and LLAMA2:** We conduct a comprehensive experiment to evaluate the performance of both LLaMA and LLaMA2 across various model scales. Table 1 presents the overall accuracy results, comparing our QRazor scheme with previous SOTA quantization methods that provide W4A4 precision. Notably, both our approach and QuaRot (Ashkboos et al., 2024) additionally offer 4-bit KV cache quantization. For our QRazor scheme, group sizes of 16 and 32 are evaluated, where

Table 2: Zero-shot accuracy of Mistral-7B & Gemma2-2B models on five common sense tasks.

| Model | #Bits | Method | Zero-shot Accuracy ↑ | | | | | |
|---|---|---|---|---|---|---|---|---|
| | | | PIQA | ARC-e | ARC-c | HellaSwag | Winogrande | Avg |
| | FP16 | Baseline | 80.74 | 81.58 | 50.16 | 61.25 | 74.27 | 69.60 |
| Mistral-7B | W4A8 $g16$ | QRazor | 79.60 | 80.35 | 47.15 | 60.51 | 72.38 | 68.00 |
| | W4A8 $g32$ | | 79.76 | 79.82 | 46.82 | 60.11 | 71.19 | 67.54 |
| | W4A8KV4 $g16$ | | 79.65 | 80.18 | 46.15 | 59.95 | 72.35 | 67.66 |
| | W4A8KV4 $g32$ | | 77.53 | 76.09 | 42.24 | 59.58 | 66.38 | 64.36 |
| | W4A4 $g16$ | QRazor | 77.75 | 77.98 | 45.48 | 57.70 | 69.77 | 65.74 |
| | W4A4 $g32$ | | 77.80 | 75.39 | 43.47 | 56.20 | 68.90 | 64.35 |
| | W4A4KV4 $g16$ | | 77.78 | 77.34 | 44.48 | 58.09 | 68.56 | 65.25 |
| | W4A4KV4 $g32$ | | 77.53 | 76.09 | 42.24 | 56.38 | 66.38 | 63.72 |
| | FP16 | Baseline | 60.17 | 37.89 | 22.41 | 35.09 | 51.07 | 41.32 |
| Gemma-2-2B | W4A8 $g16$ | QRazor | 59.09 | 38.95 | 23.41 | 34.55 | 52.80 | 41.76 |
| | W4A8 $g32$ | | 57.83 | 35.61 | 22.07 | 34.79 | 52.01 | 40.46 |
| | W4A8KV4 $g16$ | | 59.47 | 39.12 | 23.08 | 34.56 | 53.35 | 41.92 |
| | W4A8KV4 $g32$ | | 58.00 | 36.32 | 21.07 | 34.70 | 51.22 | 40.26 |
| | W4A4 $g16$ | QRazor | 57.02 | 38.77 | 23.08 | 32.82 | 51.78 | 40.69 |
| | W4A4 $g32$ | | 55.71 | 32.46 | 20.06 | 32.13 | 51.05 | 38.28 |
| | W4A4KV4 $g16$ | | 57.18 | 37.54 | 19.73 | 32.80 | 50.51 | 39.55 |
| | W4A4KV4 $g32$ | | 55.98 | 33.16 | 20.74 | 32.15 | 47.91 | 37.99 |

groups share the same flag bits. These group sizes yield the same effective bits per data as conventional group-wise quantization schemes with 128 data per group. As previously mentioned, our scheme enables direct low-precision processing, ensuring that smaller group sizes hardly degrade hardware performance, as no dequantization operations are required across groups.

While QLLM (Liu et al., 2024a) preserves the best accuracies in LLaMA models compared to previous methods, it still suffers from significant accuracy degradation, with an average drop exceeding 10%. Additionally, QLLM does not address the quantization of KV caches into a low-precision format, which represents a significant memory bottleneck during long-context LLM inference. In contrast, our QRazor scheme achieves higher accuracy in LLaMA, showing the lowest accuracy drop from the baseline. For LLaMA2 models, QuaRot is compared, showing reasonable results with a marginal accuracy drop in most tasks using W4A4KV4 precision. When comparing the methods equipped with the nearest-to-round (RTN) rounding, commonly applied for both weight and activation compression, QRazor outperforms QuaRot, exhibiting an average accuracy drop of <8% in LLaMA2 models. QuaRot suggests that applying GPTQ (Frantar et al., 2023) to the weights could further reduce the accuracy drop. In this case, results with similar or slightly better accuracy are observed compared to our method with RTN applied. Since our scheme is solely influenced by the internal data distribution and is independent of other external criteria, various optimization techniques such as GPTQ can also be applied to our 4-bit model for further enhancements. We leave these potential improvements for future work.

**Mistral-7B and Gemma2-2B:** In addition to the LLaMA models, we conducted an accuracy analysis on Mistral-7B and Gemma2-2B, achieving strong results across these models, as shown in Table 2. These findings highlight QRazor's capability to deliver reliable performance across various models and configurations.

## 5.2 ABLATION STUDIES

**W4A8 evaluation.** Although QRazor outperforms various 4-bit quantization approaches in task performance, some accuracy degradation is inevitable due to the aggressive low-precision representation. Therefore, as an extension of our scheme, we increase the number of captured salient bits to 8 bits per activation. This section demonstrates the effectiveness of the wider W4A8 version of QRazor, comparing it to QLLM, a SOTA quantization method that also supports the W4A8 setting.

Table 10 presents the accuracy results under the W4A8 precision setting. In the QRazor scheme, the W4A8 configuration demonstrates significantly higher accuracies across various tasks than W4A4. At W4A8 precision, our QRazor consistently outperforms QLLM in most cases. Additionally, we further quantize KV caches to 4 bits, denoted as W4A8KV4, resulting in minimal accuracy loss. When comparing W4A8 to W4A4 in Table 1, QRazor demonstrates less degradation in performance with reduced activation precision compared to QLLM. Notably, our SDR technique captures salient

Table 3: Zero-shot accuracy of W4A8 LLaMA2 models on five common sense tasks.

| Model | #Bits | Method | PIQA | ARC-e | ARC-c | HellaSwag | Winogrande | Avg |
|---|---|---|---|---|---|---|---|---|
| | | | | | | **Zero-shot Accuracy ↑** | | |
| LLaMA-2-7B | FP16 | - | 79.13 | 74.39 | 45.97 | 76.21 | 69.30 | 69.00 |
| | W4A8 | QLLM | 76.11 | 51.73 | 39.33 | 71.27 | 65.59 | 60.81 |
| | W4A8KV4 | QServe | 77.64 | 72.81 | 43.60 | 74.00 | 68.03 | 67.22 |
| | W4A8KV4 $g128$ | QServe | 78.07 | 73.32 | 44.80 | 74.98 | 68.59 | 67.95 |
| | W4A8 $g16$ | QRazor | 77.48 | 72.39 | 44.15 | 74.87 | 68.35 | 67.45 |
| | W4A8 $g32$ | | 77.09 | 72.91 | 44.15 | 74.50 | 68.98 | 67.53 |
| | W4A8KV4 $g16$ | | 76.55 | 73.86 | 43.14 | 73.81 | 68.59 | 67.19 |
| | W4A8KV4 $g32$ | | 76.66 | 71.75 | 43.82 | 72.97 | 68.11 | 66.66 |
| LLaMA-2-13B | FP16 | - | 80.23 | 77.82 | 48.76 | 79.39 | 72.30 | 71.70 |
| | W4A8 | QLLM | 78.67 | 57.11 | 41.89 | 75.33 | 68.75 | 64.35 |
| | W4A8KV4 | QServe | 79.71 | 75.97 | 48.38 | 77.80 | 70.96 | 70.56 |
| | W4A8KV4 $g128$ | QServe | 79.43 | 77.06 | 48.81 | 78.35 | 65.59 | 70.83 |
| | W4A8 $g16$ | QRazor | 79.33 | 76.42 | 45.14 | 78.24 | 71.74 | 70.17 |
| | W4A8 $g32$ | | 79.00 | 76.07 | 46.48 | 78.07 | 69.93 | 69.91 |
| | W4A8KV4 $g16$ | | 78.51 | 76.67 | 45.15 | 77.92 | 71.19 | 69.88 |
| | W4A8KV4 $g32$ | | 78.24 | 76.84 | 47.83 | 77.54 | 69.93 | 70.07 |

bits at the bit level, enabling finer data preservation than other methods, which proves especially advantageous in more aggressive compression settings.

Compared to another SOTA method, QServe Lin et al. (2024b), QRazor achieves nearly comparable accuracies. However, QServe employs complex rotation operations for quantizing activations, similar to Quarot, which introduces significant computational overhead. This highlights that QRazor is a more efficient technique than QServe, at the W4A8KV4 configuration.

**Effect of the SDR group size.** This section examines the impact of different group size settings in our scheme. A key feature of our approach is the use of smaller group sizes compared to other methods, as we can perform low-precision computations without the need for decompressing or dequantizing the data, thereby maintaining hardware performance. However, smaller group sizes do result in an increased size of the cached flag data. Meanwhile, other quantization approaches also require additional data, such as FP32 or FP16 scale factors per group, which must be stored and then dequantized after completing the final MAC operations within a group. This implies that each data point gains an additional 0.25 or 0.125 bits ($group\ size = 128$) beyond its quantized bit-width, a metric commonly referred to as the effective bit-width per data.

Table 4 presents the average accuracy of the LLaMA and LLaMA2 models across various tasks when QRazor is applied with different group sizes in W4A4KV4 configuration. As the group size increases, capturing salient bits becomes more challenging due to the larger number of data points sharing the same razoring point. Consequently, significant accuracy degradation is observed in some tasks when tested with a group size of 128, which shares four flag bits within the group, resulting in an effective bit-width of approximately 4.03 bits per data. Based on the average accuracy drop, a group size of 32 or smaller provides minimal accuracy loss across various models, outperforming other methods. Group sizes of 16 and 32 also yield reasonable effective bit-widths of 4.25 and 4.125, respectively, which are comparable to those of group-based quantization methods with a group size of 128. This explains why group sizes of 16 and 32 are selected as the primary configurations for comparison.

Table 4: Zero-shot accuracy and effective bit-width comparison between different group sizes of W4A4KV4 configuration.

| Group ($g$) Size | Model | Baseline | 8 | 16 | 32 | 64 | 128 |
|---|---|---|---|---|---|---|---|
| Effective Bits | | | 4.5 | 4.25 | 4.125 | 4.06 | 4.03 |
| Average Accuracy | LLaMA-1-7B | 67.28 | 64.79 | 64.18 | 61.08 | 60.06 | 53.28 |
| | LLaMA-1-13B | 70.83 | 67.96 | 67.55 | 64.74 | 63.23 | 58.53 |
| | LLaMA-2-7B | 69.00 | 64.58 | 63.65 | 61.17 | 56.91 | 47.36 |
| | LLaMA-2-13B | 71.70 | 69.18 | 68.19 | 67.07 | 65.93 | 63.76 |

## 5.3 Hardware Efficiency

The hardware efficiency of the processing unit design described in section 4.3 is evaluated by fully implementing the register-transfer-level designs in Verilog and synthesizing them with $Synopsys$ $Design\ Compiler$ using an industrial LP 65nm library. The power consumption is extracted using $PrimeTime\ PX$. As shown in Table 2, our proposed design achieves a 61.2% reduction in area compared to the 16×8 INT MAC unit, which represents the base precision arithmetic unit, and a 34% reduction compared to an 8×8 INT MAC unit, which serves as the standard precision for integer-based GEMM in most GPUs (Lin et al., 2024b). In terms of power consumption, our unit achieves reductions of 56% and 33.7% compared to the 16×8 INT MAC and 8×8 INT MAC units, respectively. Compared to the FP16 MAC unit, the area and power saving become more significant.

The significant reductions in both area and power highlight the efficiency of the proposed decompression-free method. These improvements translate into several advantages for the system, including increased computational throughput, reduced hardware resource consumption, and enhanced energy efficiency. This makes the system particularly suitable for deployment in resource-constrained environments, such as edge devices, where minimizing area and power is critical for maintaining performance.

Table 5: Comparison of power and area for MAC units

|  | FP 16×16 MAC | INT 16×8 MAC | INT 8×8 MAC | INT 4×4 Proposed |
|---|---|---|---|---|
| **Area** ($\mu m^2$) | | | | |
| Multiplier | 3042.2 | 1052.2 | 559.4 | 112 |
| Additional Shifters | 0 | 0 | 0 | 156.5 |
| Reg. + Accm. | 1127.1 | 631 | 431 | 385.3 |
| Total | 4169.3 | 1683.2 | 990.4 | 653.8 |
| **Power** ($mW$) | | | | |
| Multiplier | 0.3378 | 0.0506 | 0.023 | 0.0028 |
| Additional Shifters | 0 | 0 | 0 | 0.0067 |
| Reg. + Accm. | 0.1242 | 0.0733 | 0.0581 | 0.0451 |
| Total | 0.4620 | 0.1239 | 0.0811 | 0.0546 |

## 6 Conclusion

In this paper, we introduced QRazor, a post-training quantization (PTQ) method that enables reliable 4-bit quantization for large language models (LLMs) through two key stages: quantization and compression. Our approach effectively balances accuracy and efficiency by first quantizing weights, activations, and KV caches to 8-bit and 16-bit integer formats, followed by compressing them using the significant data razoring (SDR) technique, which preserves salient bits for low-bit computations. Experimental results on LLaMA and LLaMA2 models demonstrate QRazor's reliable performance compared to state-of-the-art methods, particularly in scenarios such as W4A4 and W4A4KV4, with minimal accuracy degradation when using smaller group sizes. Additionally, our hardware-optimized, decompression-free arithmetic unit significantly reduces area and power consumption, making QRazor ideal for deployment in resource-constrained environments. Overall, QRazor presents an efficient and scalable solution for low-bit quantization in LLMs, winning both software and hardware optimizations to enhance system performance.

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

# A APPENDIX

## A.1 PRECISION ANALYSIS OF THE QUANTIZATION STAGE

As discussed in Section 4.1, outliers pose a significant challenge in low-bit quantization of LLMs, particularly for activations, which exhibit a much broader dynamic range compared to weights or KV caches. Our observations, consistent with previous research (van Baalen et al., 2023), indicate that activations require quantization to higher precision, such as 16-bit integers, to capture this wide range without sacrificing accuracy. On the other hand, weights and KV caches can be effectively represented using 8-bit integers without significantly affecting LLM accuracy. This finding under-scores the necessity of selecting appropriate bit-widths during the quantization stage to effectively capture the full range of activation outliers, ensuring accurate representation without compromising model performance.

To validate these requirements, we evaluated LLM accuracy across three quantization scenarios: W8A8 (8-bit weights and 8-bit activations), W8A16, and W8A16KV8. Our results confirm that the W8A16 and W8A16KV8 scenarios achieve nearly identical accuracies to their FP16 counterparts across various LLaMA tasks, while W8A8 results in significant accuracy drops. These findings demonstrate that 16-bit integers are crucial for capturing the outlier-prone dynamic range of acti-vations, whereas 8-bit integers suffice for weights and KV caches. Note that our razoring scheme involves a delicate process of detecting salient bits at the bit level, but it must be applied to the base integer data with minimal information loss. Selecting the appropriate base precision during the quantization stage is, therefore, crucial to maintaining data accuracy.

Table 6: Zero-shot accuracy of different base precision settings.

| Model | #Bits | Method | Zero-shot Accuracy ↑ | | | | |
|---|---|---|---|---|---|---|---|
| | | | PIQA | ARC-e | ARC-c | HellaSwag | Winogrande |
| LLaMA-2-7B | FP16 | - | 79.13 | 74.39 | 45.97 | 76.21 | 69.30 |
| | W8A8 | | 71.89 | 65.99 | 33.11 | 65.11 | 64.33 |
| | W8A16 | Absmax Quantization | 77.69 | 74.05 | 42.43 | 75.97 | 69.30 |
| | W8A16KV8 | | 78.07 | 74.20 | 42.41 | 76.26 | 69.30 |
| LLaMA-2-13B | FP16 | - | 80.23 | 77.82 | 48.76 | 79.39 | 72.30 |
| | W8A8 | | 68.99 | 40.35 | 27.09 | 57.26 | 50.99 |
| | W8A16 | Absmax Quantization | 78.94 | 76.47 | 45.14 | 78.92 | 71.27 |
| | W8A16KV8 | | 78.73 | 76.38 | 44.81 | 79.03 | 71.51 |

## A.2 IMPORTANCE OF WEIGHT QUANTIZATION

To assess the impact of weight and activation compression on accuracy using the QRazor method, we conducted the following experiments. From the integer data quantized into our base precision of W8A16KV8, we analyzed the accuracy of W4A8, W8A8, and W4A16 configurations to examine the impact of weight compression in comparison to activation compression with group size of 8. As shown in Table 7, W8A8 demonstrates the highest accuracy among the three cases. The results suggest that compressing weight values into fewer bits is as sensitive as capturing activation outliers. When outlier characteristics are well-preserved, reducing activation bits using our SDR scheme has minimal impact on overall accuracy. In contrast, although weights are less affected by outliers, reducing their bit-width to 4 bits can introduce quantization errors, significantly impacting overall accuracy. Consequently, further optimization of weight quantization can enhance the accuracy of our 4-bit LLM, such as by applying the GPTQ (Frantar et al., 2023) technique to weight values alongside our QRazor scheme, as briefly mentioned in Section 5.2.

## A.3 SDR ENCODING: DETECTING THE RAZORING POINT WITH BITWISE OR OPERATIONS

As mentioned in Section 4.2, the primary concept of our SDR scheme is its streamlined encod-ing and decoding process compared to other methods, offering the key advantage of on-the-fly compression and decompression, particularly for activations. We have discussed the details of our decompression-free arithmetic design, which facilitates end-to-end arithmetic operations while by-

Table 7: Zero-shot accuracy comparison of QRazor between W4A8, W8A8, and W4A16

| Model | #Bits | Method | Zero-shot Accuracy ↑ | | | | |
|---|---|---|---|---|---|---|---|
| | | | PIQA | ARC-e | ARC-c | HellaSwag | Winogrande |
| LLaMA-2-7B | FP16 | Baseline | 79.13 | 74.39 | 45.97 | 76.21 | 69.30 |
| | W4A8 | | 77.04 | 75.09 | 45.48 | 74.87 | 68.90 |
| | W8A8 | QRazor | 77.75 | 75.79 | 45.82 | 75.34 | 69.14 |
| | W4A16 | | 77.20 | 74.91 | 44.82 | 74.99 | 68.98 |
| LLaMA-2-13B | FP16 | Baseline | 80.23 | 77.82 | 48.76 | 79.39 | 72.30 |
| | W4A8 | | 78.56 | 78.60 | 44.82 | 78.33 | 70.88 |
| | W8A8 | QRazor | 78.67 | 78.77 | 45.15 | 79.37 | 72.38 |
| | W4A16 | | 78.56 | 78.77 | 45.15 | 78.51 | 70.92 |

passing the decompression process. Although we have outlined the overall encoding concept of our SDR scheme, we have not yet provided detailed information regarding the hardware aspects. Figure 4 illustrates the entire SDR process, starting with detecting the razoring point, which can be implemented as simple parallel bitwise OR operations within a group. The result of the bitwise OR operation effectively identifies the MSB position within the group, and the flag bits are automatically generated based on this position. The nearest-to-round method, combined with truncation, preserves the salient bits at the target precision.

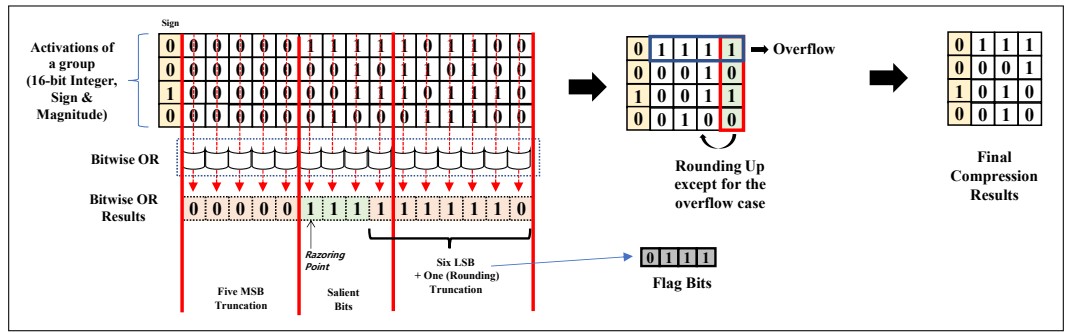

Figure 4: SDR Encoding Scheme Consisting of Bitwise OR Operations for detecting the razoring point.

### A.4 PERPLEXITY

We evaluated the performance of our QRazor scheme using perplexity metrics for W4A4, W4A8, W4A4KV4, and W4A8KV4 configurations on the LLaMA-2-7B and LLaMA-2-13B models, specifically on the Lambada-OpenAI dataset (Paperno et al., 2016). For calibration, 128 randomly selected samples from wikitext2 data has been used.

Table 8: Perplexity of different group size with Lambada-OpenAI

| Model | #Bits | Method | Group Size Perplexity ↓ | | | | | |
|---|---|---|---|---|---|---|---|---|
| | | | Baseline | 8 | 16 | 32 | 64 | 128 |
| LLaMA-2-7B | W4A8 | QRazor | 3.40 | 3.83 | 4.01 | 4.34 | 4.52 | 4.65 |
| | W4A4 | | | 4.24 | 4.47 | 5.04 | 5.83 | 7.94 |
| | W4A8KV4 | | | 4.14 | 4.31 | 4.63 | 5.04 | 6.11 |
| | W4A4KV4 | | | 4.29 | 4.98 | 5.91 | 7.87 | 19.23 |
| LLaMA-2-13B | W4A8 | QRazor | 3.04 | 3.36 | 3.28 | 3.34 | 3.38 | 3.39 |
| | W4A4 | | | 3.45 | 3.64 | 3.93 | 4.58 | 6.22 |
| | W4A8KV4 | | | 3.36 | 3.45 | 3.54 | 3.63 | 4.15 |
| | W4A4KV4 | | | 3.59 | 3.79 | 4.27 | 5.09 | 9.35 |

## A.5 OVERALL ACCURACY RESULT

We present the overall accuracy results experimented by QRazor method with LLaMA2 models in Table 9, encompassing all cases across different bit-widths and group sizes. Notably, W4A8 demonstrates negligible accuracy in all scenarios, including W4A8KV4. In W4A4 and W4A4KV4 configurations, group sizes of 16 and 32 exhibit reliable accuracy across all cases.

Table 9: Zero-shot accuracy of quantized LLaMA2 models on five common sense tasks.

| Model | #Bits | Group Size | Zero-shot Accuracy ↑ | | | | | |
| --- | --- | --- | --- | --- | --- | --- | --- | --- |
| | | | PIQA | ARC-e | ARC-c | HellaSwag | Winogrande | Avg. |
| LLaMA-2-7B | FP16 | Baseline | 79.13 | 74.39 | 45.97 | 76.21 | 69.30 | 69.00 |
| | W4A8 | 8 | 77.09 | 73.44 | 44.53 | 74.97 | 68.98 | 67.80 |
| | | 16 | 77.48 | 74.39 | 40.13 | 74.87 | 68.35 | 67.04 |
| | | 32 | 77.09 | 74.91 | 39.14 | 74.47 | 68.98 | 66.92 |
| | | 64 | 76.71 | 74.26 | 43.82 | 73.96 | 67.25 | 67.20 |
| | | 128 | 76.17 | 73.09 | 43.11 | 72.76 | 67.88 | 66.60 |
| | W4A4 | 8 | 75.30 | 72.88 | 41.14 | 73.40 | 66.30 | 65.80 |
| | | 16 | 75.84 | 72.63 | 42.47 | 72.96 | 65.67 | 65.91 |
| | | 32 | 73.67 | 70.70 | 39.46 | 71.46 | 64.09 | 63.88 |
| | | 64 | 73.99 | 69.12 | 39.46 | 69.06 | 62.98 | 62.92 |
| | | 128 | 69.91 | 63.16 | 32.78 | 63.31 | 59.27 | 57.69 |
| | W4A8KV4 | 8 | 76.55 | 73.86 | 43.14 | 73.81 | 68.59 | 67.19 |
| | | 16 | 76.66 | 71.75 | 43.82 | 72.97 | 68.11 | 66.66 |
| | | 32 | 75.46 | 71.58 | 43.38 | 66.95 | 62.04 | 63.88 |
| | | 64 | 69.97 | 61.58 | 34.78 | 61.32 | 56.91 | 56.91 |
| | | 128 | 61.81 | 52.63 | 27.76 | 43.07 | 51.54 | 47.36 |
| | W4A4KV4 | 8 | 75.35 | 70.70 | 39.80 | 71.63 | 65.43 | 64.58 |
| | | 16 | 73.39 | 70.88 | 39.80 | 70.15 | 64.01 | 63.65 |
| | | 32 | 73.23 | 67.54 | 37.46 | 67.16 | 60.46 | 61.17 |
| | | 64 | 69.97 | 61.58 | 34.78 | 61.32 | 56.91 | 56.91 |
| | | 128 | 61.81 | 52.63 | 27.76 | 43.07 | 51.54 | 47.36 |
| LLaMA-2-13B | FP16 | Baseline | 80.23 | 77.82 | 48.76 | 79.39 | 72.30 | 71.70 |
| | W4A8 | 8 | 78.78 | 78.77 | 45.82 | 78.48 | 71.74 | 70.72 |
| | | 16 | 79.33 | 76.42 | 45.14 | 78.24 | 71.74 | 70.17 |
| | | 32 | 79.00 | 76.07 | 46.48 | 78.07 | 69.93 | 69.91 |
| | | 64 | 78.67 | 75.89 | 45.48 | 77.97 | 70.56 | 69.71 |
| | | 128 | 78.62 | 75.60 | 44.14 | 77.70 | 69.93 | 69.20 |
| | W4A4 | 8 | 77.97 | 75.19 | 42.81 | 77.19 | 70.72 | 68.78 |
| | | 16 | 77.48 | 73.09 | 43.47 | 76.83 | 70.09 | 68.17 |
| | | 32 | 76.82 | 71.86 | 41.80 | 76.30 | 68.59 | 67.07 |
| | | 64 | 76.12 | 70.98 | 41.36 | 74.74 | 67.01 | 66.04 |
| | | 128 | 74.97 | 70.11 | 38.54 | 73.12 | 62.04 | 63.76 |
| | W4A8KV4 | 8 | 79.05 | 77.19 | 49.16 | 78.32 | 71.43 | 71.03 |
| | | 16 | 78.51 | 76.67 | 45.15 | 77.92 | 71.19 | 69.89 |
| | | 32 | 78.24 | 76.84 | 47.83 | 77.54 | 69.93 | 70.08 |
| | | 64 | 78.02 | 76.14 | 47.81 | 76.71 | 69.77 | 69.69 |
| | | 128 | 77.20 | 73.33 | 44.15 | 74.66 | 68.98 | 67.67 |
| | W4A4KV4 | 8 | 76.77 | 75.61 | 46.49 | 76.55 | 71.11 | 69.31 |
| | | 16 | 76.55 | 75.44 | 44.15 | 75.38 | 68.43 | 67.99 |
| | | 32 | 75.35 | 71.75 | 44.28 | 73.61 | 66.77 | 66.35 |
| | | 64 | 74.59 | 71.40 | 42.12 | 70.84 | 62.19 | 64.23 |
| | | 128 | 68.82 | 62.81 | 32.78 | 58.64 | 55.17 | 55.64 |

## A.6  FLOPS & OPS COMPARISON RESULT

To evaluate the computational efficiency of our proposed quantization scheme relative to the approach outlined in the QuaRot paper, we conducted a comparative analysis of the FLOPs and OPs involved in the attention layer of a transformer architecture. QuaRot employs two primary operations—Hadamard product and rotation operation—for outlier-aware quantization. These operations effectively reduce the impact of outliers during quantization by transforming datasets into lower-variance distributions. This process enables efficient mapping of data to 4-bit representations using conventional per-tensor or per-channel quantization techniques.

However, the Hadamard product and rotation operations introduce substantial computational overhead during both quantization and dequantization. Notably, the dequantization process in QuaRot incurs a significant number of additional FLOPs, as it involves dequantization computations for weights as well. Consequently, while QuaRot is effective for quantization, its reliance on FLOPs-intensive operations reduces computational efficiency when compared to other methods with similar OPs.

In contrast, our proposed quantization scheme simplifies the dequantization process by utilizing shifting operations immediately after INT matrix-multiplication(matmul) to perform decompression. This design choice results in a 35% increase in OPs relative to QuaRot. However, it dramatically reduces the FLOPs required, demanding only 3% of the FLOPs utilized by QuaRot. This substantial reduction in FLOPs contributes to increased throughput and lower power consumption, underscoring the computational efficiency and practical benefits of our method compared to QuaRot.

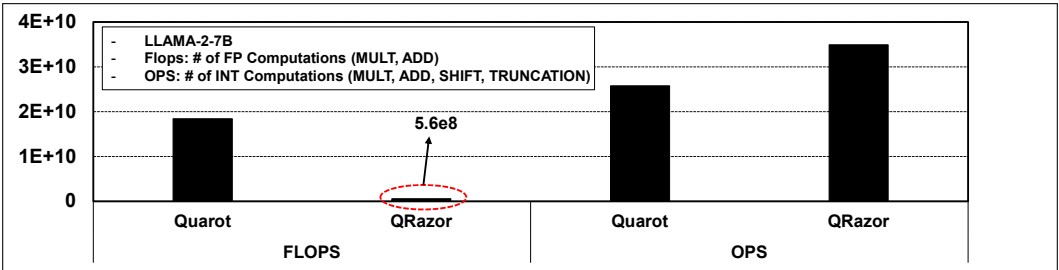

Figure 5: FLOPs & OPs comparsion with QuaRot in Attention Layer

## A.7  ACCURACY COMPARISON WITH SOTA WORK

Here, we conduct accuracy comparison with 4bit quantization SOTA works with Mistral-7B model. In all configurations, our method achieved highest performance by average accuaracy drop of less than 6% in five common sense tasks.

Table 10: Zero-shot accuracy of Mistral-7B models on five common sense tasks.

| Model | #Bits | Method | Zero-shot Accuracy ↑ | | | | | |
| | | | PIQA | ARC-e | ARC-c | HellaSwag | Winogrande | Avg |
|---|---|---|---|---|---|---|---|---|
| Mistral-7B | FP16 | Baseline | 80.74 | 81.58 | 50.16 | 61.25 | 74.27 | 69.60 |
| | W4A4 | SmoothQuant | 57.94 | 35.14 | 21.75 | 30.51 | 48.30 | 38.73 |
| | | OS+ | 66.70 | 56.73 | 30.20 | 42.39 | 52.01 | 49.61 |
| | | AWQ | 66.26 | 54.16 | 30.80 | 43.35 | 53.67 | 49.67 |
| | | TesseraQ* | 72.19 | 65.90 | 33.78 | 49.02 | 57.61 | 55.71 |
| | W4A4 $g16$ | QRazor | 77.75 | 77.98 | 45.48 | 57.70 | 69.77 | 65.74 |
| | W4A4 $g32$ | | 77.80 | 75.39 | 43.47 | 56.20 | 68.90 | 64.35 |
| | W4A4KV4 $g16$ | QRazor | 77.78 | 77.34 | 44.48 | 58.09 | 68.56 | 65.25 |
| | W4A4KV4 $g32$ | | 77.53 | 76.09 | 42.24 | 56.38 | 66.38 | 63.72 |

## A.8 QRAZOR QUANTIZATION FLOW IN TRANSFORMERS

To illustrate the quantization flow of the QRazor method, we provide a detailed depiction of the Transformer layer. In comparison to methods like QuaRot(Ashkboos et al., 2024) and Spin-Quant(Liu et al., 2024b), our approach uniquely quantizes the Query, enabling a decompression-free INT4 matrix multiplication for the $Q \cdot K^\top$ operation within the Multi-Head Attention layer. This capability significantly reduces the computational burden across the Transformer layer and improves overall throughput.

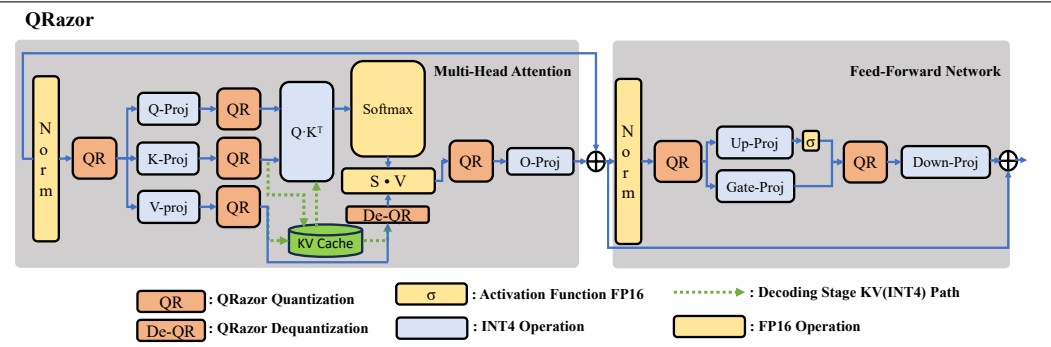

Figure 6: QRazor Quantiztaion Flow in Attention & FFN Layer