# OpenReview forum: "QRazor: Reliable and Effortless 4-bit LLM Quantization by Significant Data Razoring"
_ICLR.cc/2025/Conference — Submitted to ICLR 2025_

### Official Review · Reviewer_WgNW · 2024-10-22

**Soundness:** 2
**Presentation:** 1
**Contribution:** 2
**Rating:** 3
**Confidence:** 5

**Summary:**

The paper proposes QRazor. A fine-grained quantization techniques that shows an alternate way of doing max scaling at the group-level in terms of bit positioning and bitwise operations. The approach is however identical to fine-grained dynamic max scaling that has been massively explored before.

**Strengths:**

Unfortunately, there is not much that can be said here.

**Weaknesses:**

There are several weaknesses - please see the questions. To summarize, there are issues with respect to the novelty of the technique. QRazor is just doing fine-grained dynamic max scaled quantization, but it is disguised as a bunch of bitwise operations. In addition, the empirical evaluation is weak, focusing only on multiple choice questions.

**Questions:**

- Why is min-max scaling introduced if it is not used in the work. There's also no reference accompanying it. Section can be made shorter if we only introduce absolute max-scaling as the PTQ recipe employed (it probably won't even need to be a section of its own).
- Figure 1 is very hard to read without significant zooming and horizontal scrolling. The caption is also not helpful in understanding how the QRazor approach works.
- What does "2s complement except the sign bit for negative activations" mean? Is that referring to 1s complement or sign-magnitude?
- In line 191-192 the authors argue that a coarser grained scaling is favorable for activation quantization since it has to be done online. This is wrong. Since activation quantization is done online, its scaling and quantization should occur in the epilogue of the prior operation (either a layernorm or an activation function). Fine-grained scaling and quantization can be fused in cuda with said prior ops whereas coarse grained scaling would require a reduction operation (such as max) over the whole tensor (or channel) which would be slow.
- The proposed SDR technique is exactly the same as a dynamic per-group max. It just provides a hardware method to do it when the groups are coming from high-precision integers representations.
- In the results section, in Table 1: why is there a duplication of FP16 baseline for each model? And why are the duplicated baselines not having similar accuracy?
- Since W4A8 is highlighted, can we have a comparison to AWQ?
Can the authors add perplexity of the models in addition to scores on multiple choice questions (which can be noisy)? I suggest using Wikitext-103 test set.

---

> ### Author Response · Authors · 2024-11-24
> **W1**
>
> Thank you for your careful review and recommendation! We would like to answer them one at a time. The revised parts of the paper are colored in blue. As we answer to your questions, we will cite the table or figure numbers for your better understanding. Please refer to our citations for more information of our answers. We have incorporated your suggestion into the revised paper, which dedicated to improvement of the paper's quality which we are sincerely grateful for.
>
> **W1**: *There are several weaknesses - please see the questions. To summarize, there are issues with respect to the novelty of the technique. QRazor is just doing fine-grained dynamic max scaled quantization, but it is disguised as a bunch of bitwise operations. In addition, the empirical evaluation is weak, focusing only on multiple choice questions.*
>
> **A1**: We appreciate your comments. In Section 4.2 of our revised paper, we included the following explanation.
>
> It is important to note that in our compression, the truncation level dynamically varies for each group during runtime. Due to this characteristic, one might draw a comparison between our compression and dynamic max-scaled quantization (DMQ). However, our compression technique fundamentally differs from DMQ in the following aspects:
>
> ------------------------------------------------------------------------------------------------------------------------------------------
> 1. Eliminating Absolute Max Computation: Instead of determining the absolute maximum value within a group, QRazor detects only the leading '1'. While the absolute maximum inherently contains the leading '1', multiple parameters may share the same leading '1' position. In DMQ, identifying the absolute maximum which is expressed in floating point data format, is essential for computing the scaling factor. In contrast, our compression bypasses this step entirely, significantly reducing computational complexity.
> --------------------------------------------------------------------------------------------------------------------------------------------
> 2. Lightweight Compression and Decompression: Per-group DMQ typically involves group-level quantization and dequantization operations, which rely on arithmetic computations such as multiplication and, in some cases, division. In contrast, our compression and decompression rely on bit-level truncation and shifting, which are inherently simpler and far less resource-intensive.
> --------------------------------------------------------------------------------------------------------------------------------------------
> To conclude, while DMQ and our compression share the goal of dynamically reducing precision, their underlying methodologies are fundamentally different. As outlined above, QRazor’s compression incurs substantially lower computational overhead than DMQ. Moreover, QRazor’s simpler operations require fewer hardware resources, enabling more efficient implementation in dedicated arithmetic units, as further discussed in the following section.

---

> > ### Comment · Reviewer_WgNW · 2024-11-24
> > **This is exactly what MXFP and VS-quant do**
> >
> > Thank you for the response. The described procedure is exactly what prior works such as MXFP and VS-quant do. They use a power-of-2 scale factor per block which is equivalent to finding the leading 1.
> >
> > Rouhani, Bita Darvish, et al. "Microscaling data formats for deep learning." arXiv preprint arXiv:2310.10537 (2023).
> >
> > Dai, Steve, et al. "Vs-quant: Per-vector scaled quantization for accurate low-precision neural network inference." arXiv preprint arXiv:2102.04503 (2021).

---

> ### Author Response · Authors · 2024-11-24
> **Q1~Q3**
>
> **Q1**: *Why is min-max scaling introduced if it is not used in the work. There's also no reference accompanying it. Section can be made shorter if we only introduce absolute max-scaling as the PTQ recipe employed (it probably won't even need to be a section of its own).*
>
> **A2**: Thank you for your comment. We agreed to your suggestion and removed the following part from our revised paper.
>
> -------------------------------------------------------------------------------------------------------------------------------------------
>
> **Q2**:  *Figure 1 is very hard to read without significant zooming and horizontal scrolling. The caption is also not helpful in understanding how the QRazor approach works.*
>
> **A3**: We apologize for inconvenience of Figure 1's visibility. We have fully redraw the figure in our revised paper. Thank you for your comments.
>
> -------------------------------------------------------------------------------------------------------------------------------------------
>
> **Q3**: *What does "2s complement except the sign bit for negative activations" mean? Is that referring to 1s complement or sign-magnitude?*
>
> **A4**: We are sorry that we made confusion to you. It is sign and magnitude conversion. In the revised paper, we clarify it through Figure 1.

---

> ### Author Response · Authors · 2024-11-24
> **Q4**
>
> **Q4**: *In line 191-192 the authors argue that a coarser grained scaling is favorable for activation quantization since it has to be done online. This is wrong. Since activation quantization is done online, its scaling and quantization should occur in the epilogue of the prior operation (either a layernorm or an activation function). Fine-grained scaling and quantization can be fused in cuda with said prior ops whereas coarse grained scaling would require a reduction operation (such as max) over the whole tensor (or channel) which would be slow.*
>
> **A5**: Before presenting our work, we need to clarify one aspect. As discussed in this paper, our QRazor scheme operates in two stages: quantization and compression. **In the quantization stage, a traditional static PTQ approach is employed.**
>
> In the quantization stage, activations and KV-caches utilize per-tensor and per-head scaling, respectively, while weights employ per-channel scaling, all with static scaling parameters. Per-tensor, per-head, and per-channel scaling are considered more efficient than per-group scaling on GPUs due to their simpler implementation and reduced memory overhead. This is supported by the milestone paper, “SmoothQuant: Accurate and Efficient Post-Training Quantization for Large Language Models”[1].
>
> We acknowledge that our explanation of coarse granularity scaling and static scaling parameters may have been unclear, potentially causing confusion among reviewers. Specifically, we did not explicitly clarify that all scaling parameters in the quantization stage are static, which might have led to misunderstandings about the computational trade-offs. We apologize for this oversight and clarify it in our revision.
>
>
> [1] Xiao et al. SmoothQuant: Accuracte and Efficient Post-Training Quantization for Large Language Models. ArXiv, abs/2211.10438

---

> ### Author Response · Authors · 2024-11-24
> **Q5**
>
> **Q5**: *The proposed SDR technique is exactly the same as a dynamic per-group max. It just provides a hardware method to do it when the groups are coming from high-precision integers representations.*
>
> **A6**: We already have answered for Q5 in the previous comment. Please, refer to the above discussion.

---

> ### Author Response · Authors · 2024-11-24
> **Q6**
>
> **Q6**: *In the results section, in Table 1: why is there a duplication of FP16 baseline for each model? And why are the duplicated baselines not having similar accuracy?*
>
> **A7**: Firstly, we apologize for the confusion. In the initial submission, we could not directly compare QRazor with Quarot due to transformers and datasets version compatibility issues. Specifically, we conducted our experiments using the opensource version of HuggingFace (Ejafa/llama_7B, NousResearch/Llama-2-7b-hf, etc) which was optimized for huggingface benchmark models, while Quarot used the original Meta-llama2 version. Consequently, we relied on indirect comparisons, which may have caused confusion among reviewers.
>
> In this revision, we have fully updated the accuracy and perplexity results to use the same baseline, eliminating any confusion during comparisons and enabling direct evaluations over SOTA works such as Quarot.

---

> ### Author Response · Authors · 2024-11-24
> **Q7**
>
> **Q7**: *Since W4A8 is highlighted, can we have a comparison to AWQ?*
>
> **A8**: AWQ performs weight-only quantization to W4A16, where activations remain in FP16. Due to this reliance on FP16 activations, AWQ is not a suitable comparison target for QRazor. We assume you are referring to Qserve[1], another milestone work from the same research group, which performs W4A8KV4 quantization.
>
> In **Table 3** of our revised paper, we compare QRazor’s W4A8KV4 configuration to that of Qserve. The accuracies achieved by QRazor and Qserve are highly comparable. However, unlike Qserve, which does not support W4A4 scenarios, our QRazor demonstrates strong performance even under this more aggressive quantization setting. Further, QServe employs complex Hadamard rotation operations for quantizing activations, similar to Quarot, which introduces significant computational overhead. Please, refer to the answer for reviewer xoaX’s comments.
>
>
> [1] Lin et al.Qserve: W4A8KV4 Quantization and System Co-design for Efficient LLM Serving. ArXiv abs/2405.04532

---

> ### Author Response · Authors · 2024-11-24
> **Q8**
>
> **Q8**: Can the authors add perplexity of the models in addition to scores on multiple choice questions (which can be noisy)? I suggest using Wikitext-103 test set.
>
> **A9**: We have conducted experiments on the Wikitext-103 test set from EleutherAI/wikitext_document_level. The results are presented in **Table.R1** below.
>
>
> | **Model**     | **#Bits**         | **Method**       | **Wikitext103** |
> |:---------------:|:-------------------:|:------------------:|:----------:|
> |               |  FP16     | Baseline | 9.39|
> |               |    W4A4(g16)    |          | 11.75    |
> |      **Llama-7B**    |    W4A4(g32)   |  QRazor              | 13.14 |
> |               |    W4A4KV4(g16)   |               |12.38|
> |               |    W4A4KV4(g32)   |               | 13.22|
> |---------------|-------------------|------------------|----------|
> |               |  FP16     | Baseline                 | 8.41|
> |               |    W4A4(g16)    |                     | 9.86|
> |     **Llama-13B**   |    W4A4(g32)   | QRazor              | 10.47|
> |               |    W4A4KV4(g16)   |               | 10.18|
> |               |    W4A4KV4(g32)   |               | 10.95|
> |---------------|-------------------|------------------|----------|
> |               |  FP16     | Baseline                 | 8.73|
> |               |    W4A4(g16)    |                     | 11.26|
> |     **Llama-2-7B**   |    W4A4(g32)   | QRazor              | 12.02|
> |               |    W4A4KV4(g16)   |               | 12.07|
> |               |    W4A4KV4(g32)   |               | 12.79|
> |---------------|-------------------|------------------|----------|
> |               |  FP16     | Baseline                 | 7.75|
> |               |    W4A4(g16)    |                     | 8.89|
> |     **Llama-2-13B**   |    W4A4(g32)   | QRazor              | 9.43|
> |               |    W4A4KV4(g16)   |               | 9.31|
> |               |    W4A4KV4(g32)   |               | 10.22|

---

> ### Author Response · Authors · 2024-11-24
> **Reply to: (This is exactly what MXFP and VS-quant do)**
>
> We appreciate your comments regarding our work.
>
>  We are fully aware of the contributions of the reference papers; however, we would like to note that the methods employed differ significantly from our approach, aside from the underlying concept. Unlike typical per-group quantization using a scaling factor as 'power of 2', our method avoids the heavy computational burden of dynamic quantization to realize on-the-fly 'per-small group' INT4 representation using our proposed compression technique. Our approach only leverages parallelizable bitwise and logical shift operations for this step, where details are provided below.
>
> **Regarding MX format**
> - The MX Format is a de-facto standard for aggressive quantization, using a shared power-of-two scaling parameter per group. We consider the MX format conceptually similar to the final quantized outputs of our QRazor, as the flag bit in QRazor is analogous to the power-of-two scaling in MX Format. However, the two approaches have distinct differences, as outlined below.
>
>      **1. Quantization Approach**
>      -  MX Format is designed to directly quantize high-precision floating-point values (e.g., FP16, BFP16, FP32) into low-precision MX representations. The format itself **does not specify a single quantization method** providing mx compliant format such as FP4, FP6, FP8 and INT8.
>
>      -  In contrast, QRazor compresses high-precision integers to 4 bits using power-of-two scaling, determined by detecting the leading one position via a bitwise OR operation. As detailed in our paper, this process is computationally lightweight
>
>      **2. Handling Overflows**
>      -  When MX Format uses a 4-bit shared exponent, there is a non-negligible risk of overflow, requiring specialized techniques to manage these cases.
>      -  QRazor, on the other hand, inherently avoids overflow due to its compression methodology, eliminating the need for exception handling. This reduces computational overhead compared to MX Format conversions.
>
>
> **Regarding VS-Quant**
> - VS-Quant employs a two-stage quantization method and ultimately represents data as integers with a shared integer scaling parameter, which shares some conceptual similarities with our QRazor. However, the two approaches exhibit significant differences, as outlined below:
>
>      **1. Quantization Approach**
>      -  In VS-Quant, floating-point parameters are first quantized to integers using floating-point scaling parameters. In the second stage, the floating-point scaling parameters themselves are quantized to integers using a shared floating-point scaling parameter.
>
>      -  In QRazor, floating-point parameters are first quantized to high-precision integers using a floating-point scaling parameter. At the second stage, these high-precision integers are compressed into 4-bit integers with a shared 4-bit flag, which is efficiently computed using a bitwise OR operation, as described in our paper.
>
>      **2. Compuational Overhead**
>      -  While the first quantization stage is computationally similar in both methods, QRazor’s second-stage compression involves significantly lower computational overhead than VS-Quant, as it bypasses the need to evaluate group maximums.
>
>      **3. Key Differences in Quantization Methodology**
>      -  QRazor compresses high-precision integers into 4-bit integers with a shared 4-bit flag at the second stage, whereas VS-Quant quantizes floating-point scaling parameters not the parameters itself, into integers at the second stage.
>
>      -  QRazor employs static per-tensor scaling in the first stage, while VS-Quant uses dynamic scaling, which, as noted by reviewers, can impose substantial computational penalties, particularly in terms of speed.
>
>      **4. Accuracy Validation**
>     -  To our knowledge, VS-Quant has not yet demonstrated accuracy levels comparable to QRazor in all 4-bit LLM quantization scenarios, where the performance of VS-Quant in this context remains unvalidated.
>
>      **5. Power-of-two scale**
>     - Based on our analysis, it appears that VS-Quant **does not utilize power-of-two scale factors** when quantizing either the values or the scale parameters in any of the evaluated cases.

---

> > ### Comment · Reviewer_WgNW · 2024-11-24
> > **Please check your implementation**
> >
> > Thank you for reporting Wikitext-103 perplexity. I am afraid that the results are very inconsistent with what is reported in the literature. In the llama family, 7B and 13B models are known to have perplexities on this dataset around 5 and 4.5, respectively. This is very different from the reported numbers in the rebuttal. I suspect something went wrong in the implementation, possibly in the tokenization stage.

---

> > > ### Author Response · Authors · 2024-12-02
> > > **Response to Wikitext-103 perplexity**
> > >
> > > We appreciate your valuable comments. The discrepancies arise due to the default setting of LLama models for the Wikitext-103 dataset in our benchmark, which uses word-level perplexity as the default metric. A milestone paper on evaluation of language models **"Lessons from the Trenches on Reproducible Evaluation of Language Models"**[1] highlighted that word-level perplexity is a more suitable metric for comparing model quality. Therefore, we adhered to this default setting in our initial results. However, the scores mentioned by the reviewer WgNW appear to be based on token-level perplexity. To address this, we adjusted our settings accordingly and provide the updated results below.
> > >
> > > | **Model**     | **#Bits**         | **Method** | **Wikitext103** |
> > > |:---------------:|:-------------------|:------------------|:----------:|
> > > ||  FP16     | Baseline |5.52|
> > > ||    W4A8(g16)    || 6.10|
> > > ||    W4A8(g32)    || 6.26|
> > > ||    W4A4(g16)   || 6.59|
> > > |**Llama-7B**|    W4A4(g32)    |QRazor| 7.09|
> > > ||    W4A8KV4(g16)   || 6.25|
> > > ||    W4A8KV4(g32)   || 6.49|
> > > ||    W4A4KV4(g16)   || 6.83|
> > > ||    W4A4KV4(g32)   || 7.50|
> > > |---------------|-------------------|------------------|----------|-----------|-----------|---------------|----------------|----------|
> > > ||  FP16     | Baseline |5.11|
> > > ||    W4A8(g16)    || 5.31|
> > > ||    W4A8(g32)    || 5.40|
> > > ||    W4A4(g16)   || 5.67|
> > > |**Llama-13B**|    W4A4(g32)    |QRazor| 5.90|
> > > ||    W4A8KV4(g16)   || 5.39|
> > > ||    W4A8KV4(g32)   || 5.50|
> > > ||    W4A4KV4(g16)   || 5.77|
> > > ||    W4A4KV4(g32)   || 6.08|
> > > |---------------|-------------------|------------------|----------|-----------|-----------|---------------|----------------|----------|
> > > ||  FP16     | Baseline |5.43|
> > > ||    W4A8(g16)    || 5.86|
> > > ||    W4A8(g32)    || 6.02|
> > > ||    W4A4(g16)   || 6.49|
> > > |**Llama-2-7B**|    W4A4(g32)    |QRazor| 7.05|
> > > ||    W4A8KV4(g16)   || 6.14|
> > > ||    W4A8KV4(g32)   || 6.35|
> > > ||    W4A4KV4(g16)   || 7.57|
> > > ||    W4A4KV4(g32)   || 8.26|
> > > |---------------|-------------------|------------------|----------|-----------|-----------|---------------|----------------|----------|
> > > ||  FP16     | Baseline |4.93|
> > > ||    W4A8(g16)    || 5.12|
> > > ||    W4A8(g32)    || 5.17|
> > > ||    W4A4(g16)   || 5.37|
> > > |**Llama-2-13B**|    W4A4(g32)    |QRazor| 5.56|
> > > ||    W4A8KV4(g16)   || 5.24|
> > > ||    W4A8KV4(g32)   || 5.34|
> > > ||    W4A4KV4(g16)   || 5.54|
> > > ||    W4A4KV4(g32)   || 5.84|
> > >
> > > Consistent with the word-level perplexity findings, our QRazor demonstrates competitive perplexity scores compared to the base FP16 model under token-level settings as well, further validating the effectiveness of our QRazor scheme.
> > >
> > > We sincerely thank you for your insightful comments and hope this additional clarification and updated results address your concerns effectively.
> > >
> > >
> > > [1] Biderman et al. Lessons from the Trenches on Reproducible Evaluation of Language Models. ArXiv, abs/2405.14782

---

> ### Author Response · Authors · 2024-12-02
> **Official Comment by Authors**
>
> Dear Reviewer WgNW,
>
> For your queries, we carefully prepared the answers. We look forward to you considering raising your score if our response effectively addressed your question.
>
> Thank you and wish you have a good end of the year!
>
> Authors

---

> > ### Comment · Reviewer_WgNW · 2024-12-02
> > **Thank you, but I am still not convinced**
> >
> > I thank the authors for following up. The concerns I still have are as follows:
> >
> > 1. Wikitext-103 Perplexity results on the common opensource Llama2-7B are still off compared to what is usually reported in the literature. This is likely something related to the baseline implementation the authors are using, and casts doubts on the results.
> >
> > 2. More importantly, the authors give a very weak rebuttal on the difference between their implementation and existing fine-grained quantization techniques. I think the authors consider that VS-Quant's search of group-wise maximum is a computationally expensive operation as opposed to the detection of the leading 1 in the bit representation. This is where I strongly disagree. In a GPU implementation, or similar implementation of custom hardware with parallel processing elements, both require a small loop over the local data along with minimal computation to obtain appropriate fine-grained scaling, followed by fine-grained quantization. Such computational steps are extremely similar, and even if the semantics are minimally different, these steps are never a major contributor to hardware performance when compared to memory access from global memory and math on tensor core or similar engine.
> >
> > I also wish the authors a good end to their year.

---

> > > ### Author Response · Authors · 2024-12-03
> > > **Replies for your concerns**
> > >
> > > Thank you for your reviews. For better understanding of our work, we would like to reply to each questions.
> > >
> > >
> > > - We utilized a credible code, lm-evaluation-harness from EleutherAI, to estimate the perplexity of Wikitext-103. This code ensures reliable and reproducible results. The code and additional details can be accessed through the following link: [https://github.com/EleutherAI/lm-evaluation-harness]. We encourage you to refer to it for verification and further information.
> > >
> > > - We believe that Wikitext-2 is more commonly utilized in state-of-the-art (SOTA) works to evaluate the quality of their quantization schemes. Thus, we also conducted our evaluation using this dataset. Notably, the baseline perplexity aligns with that reported in SOTA works, thereby addressing the reviewer’s concerns and eliminating any doubts regarding the validity of our comparisons.
> > >
> > > | **Model**     | **#Bits**         | **Method** | **Wikitext2** |
> > > |:---------------:|:-------------------|:------------------|:----------:|
> > > ||  FP16     | Baseline |5.68|
> > > |**Llama-7B**|    W4A4KV4(g16)    |QRazor| 6.83|
> > > ||    W4A4KV4(g32)   || 7.52|
> > > |---------------|-------------------|------------------|----------|-----------|-----------|---------------|----------------|----------|
> > > ||  FP16     | Baseline |5.09|
> > > |**Llama-13B**|    W4A4KV4(g16)    |QRazor| 5.79|
> > > ||    W4A4KV4(g32)   || 6.11|
> > > |---------------|-------------------|------------------|----------|-----------|-----------|---------------|----------------|----------|
> > > ||  FP16     | Baseline |5.47|
> > > |**Llama-2-7B**|    W4A4KV4(g16)    |QRazor| 7.62|
> > > ||    W4A4KV4(g32)   || 8.14|
> > > |---------------|-------------------|------------------|----------|-----------|-----------|---------------|----------------|----------|
> > > ||  FP16     | Baseline |4.88|
> > > |**Llama-2-13B**|    W4A4KV4(g16)    |QRazor| 5.49|
> > > ||    W4A4KV4(g32)   || 5.79|

---

> > > ### Author Response · Authors · 2024-12-03
> > > **Replies for your concerns**
> > >
> > > **2. Regarding the other point**
> > >
> > >
> > > - It is noteworthy that no reported 4-bit integer LLM schemes utilizing fine-grained dynamic quantization have achieved the level of accuracy demonstrated by QRazor. As detailed in our work, QRazor attains this performance without requiring any additional tuning effort beyond static per-tensor/channel/head max scaling. Achieving such high accuracy with minimal effort—a significant challenge in this domain—underscores the robustness and efficiency of our approach.
> > > - Regarding fairness in memory usage, which the reviewer rightly emphasized, we consider scenarios with the same effective bitwidth, including scaling parameter memory. QRazor with G32/G16 achieves the same bitwidth as configurations where the fine-grained group size with a dynamic FP16 scaling parameter is set to 128/64. For computational fairness on tensor cores or similar engines, where 4-bit integer arithmetic is employed for matrix multiplication, no existing works deliver QRazor-level accuracy under these conditions.
> > > - The core contributions of QRazor are its lightweight compression, high accuracy, and minimal tuning effort. This combination of attributes is precisely why the title of this paper is “QRazor: Reliable and Effortless 4-bit LLM Quantization by Significant Data Razoring.” Unfortunately, the review primarily focuses on lightweight compression, overlooking the equally critical aspects of high accuracy and ease of tuning that define QRazor’s unique value proposition.
> > > - The hardware performance benefits of QRazor could indeed vary depending on the specific hardware platform and operating scenarios, which we have not explicitly compared despite QRazor's potential advantages. However, it is important to emphasize that the energy efficiency achieved by QRazor is significantly superior to conventional dynamic fine-grained scaling. This aspect is especially critical in dedicated accelerators designed for 4-bit LLMs, where energy efficiency plays a more pivotal role than computational performance alone.

---

> ### Author Response · Authors · 2024-12-03
> **Replies for your concerns**
>
> Reviewer WgNW's points can be categorized as two problems as follows.
>
>
> -	Similarity to VS-Quant
>
>
> -	Conventional Fine-grained Scaling Computation is not so significant compared to global memory and math on tensor core or similar engine.
>
>
> **1. Regarding Similarity to VS-Quant**
> - We firmly assert that VS-Quant is fundamentally different from QRazor, even in the final quantization data format. Notably, VS-Quant does not utilize power-of-two scaling. Moreover, the quantization methodologies employed by VS-Quant and QRazor are entirely distinct.
> - Although VS-Quant has not yet explored the quantization of recent LLMs like LLaMA, it shows significant accuracy degradation (3.46%) even with BERT-base at W4A8. In contrast, QRazor demonstrates negligible accuracy loss under similar conditions compared to FP16, highlighting its robustness and efficiency.
> - Qualitatively, the quantization of scaling parameters in VS-Quant introduces shared quantization errors, which can adversely impact accuracy by propagating these errors across the group, thereby degrading overall performance.
> - In the following table, we assume that VS-Quant quantizes parameters to INT4 (for fairness). Our discussion primarily focuses on the quantization of activations, as weights are quantized offline.
>
> || QRazor  | VS-Quant |
> |:-------------------:|:-------------------------------------------------------------------------------------------|:---------------------------------------------|
> |First Stage|  1. **FP16 parameters** → INT16s * one FP16 scaling parameter | 1. **FP16 parameters** → INT4s * one FP16 scaling parameter |
> ||2. Static Scaling|2. Dynamic Scaling|
> ||3. Scaling Granularity: Tensor  |3. Scaling Granularity: Large-size Vector|
> |--------------|---------------------------------------------------------------|--------------------------------------------------------------------------------|
> |Second Stage | 1. INT16s → INT4s * 2^( 4-bit Flag): The flag is shared by a group |1. **FP16 scaling parameter (Coarse-Grained)** → INT4s * FP16 scaling parameter (Fine-grained)|
> ||  2. Dynamic Compression  |2. Dynamic Scaling |
> || 3. Compression Granularity: Group (Fine-grained) |3. Scaling Granularity: Small-size Vector (Fine-grained)|
> |--------------|---------------------------------------------------------------|--------------------------------------------------------------------------------|
>
> - Finally, the quantization output is significantly different as follows.
>
>
>   **a.** QRazor:  FP16 parameters → INT4s * 2^( 4-bit Flag shared by a fine-grained group) * one FP16 scaling parameter (shared
>   by a tensor)
>
>
>   **b.** VS-Quant: FP16 parameters → INT4s * INT4 (Shared by a fine-grained vector) * one FP16 scaling parameter (shared by a
>   large-size vector).
>
> - Unlike QRazor, VS-Quant employs dynamic scaling for both the first and second stages. For a fair comparison in terms of memory requirement, let us assume that the large vector used in VS-Quant is treated as a tensor. In this scenario, VS-Quant must perform tensor-wise dynamic scaling, which, as the reviewer has already noted, is extremely slow and computationally intensive. Such a thing is mentioned in Section 4 of the VS-Quant paper.

---

### Official Review · Reviewer_XPyT · 2024-11-03

**Soundness:** 3
**Presentation:** 3
**Contribution:** 3
**Rating:** 6
**Confidence:** 4

**Summary:**

The paper presents Qrazor, a post training quantization technique pushing toward 4-bit quantization for weights, activations, as well as KV cache values (optionally). Quantization is block based and performed in two stages. First, the blocks of the tensors are quantized to high-bit-width integers (16 bits), then a "razoring" technique is applied to reach the final target quantization bit-width: after conversion to sign-magnitude, the most significant non-zero bit is identified among all pre-quantized (16bit) elements within the target block. All the zero bits at more significant digits are removed from all the elements, then the next 4-bit in order of decreasing significance are retained, with rounding applied to the last bit (flooring as special case if rounding would cause overflow in the 4-bit representation.   The authors also study the powe advantage obtainable by performing the tensor operations at low precision instread of pre-expanding the quanzized numbers. The compare against previously published 4-bit quantization techniques.

**Strengths:**

The explanation is clear, the technique presented is  simple and straightforward: a combination of known techniques is applied leading to the method presented.
Authors provide also a (simplified) estimation of power savings enabled by the technique with a dedicated unit that computes directly in low pecision formats.
Comparison with respect to state-of-the-art competitive methods is presented.

**Weaknesses:**

The analysis of outlayers is quite superficial. It's unclear why it's not always the case that the initial truncation does not always lead a 1 in the most significant bit: if the Max value of the block is used for quantization to 16-bit, it should be the case, however if a max on a much larger block is used, then it can very well be the case that the max will squeeze almost all the bits of an entire block to zero). Since the max will tend to be the outlayer, then the loss of precision on all the numbers in a block could be massive.
There is no discussion on implementation on existing hardware
The hardware idea proposed is discussed in isolation, leading to a inflation on the potential power advantages, since it neglects lots of HW that would not change (e.g. all the memory hieararchy).
Comparison wrt SoA shows that improvements over fine-tuned previous work does not show super-impressive advantages.

**Questions:**

It would be very good to see for each block which bit would be non-zero on average, i.e. on average how many leading bits can be truncated, also it would be good to see the percentage of zerooed elements per block after the aggressive quantization step.

---

> ### Author Response · Authors · 2024-11-24
> **W1**
>
> Thank you for your careful review and recommendation! We would like to answer them one at a time. The revised parts of the paper are colored in blue. As we answer to your questions, we will cite the table or figure numbers for your better understanding. Please refer to our citations for more information of our answers. We have incorporated your suggestion into the revised paper, which dedicated to improvement of the paper's quality which we are sincerely grateful for.
>
> **W1**: *The analysis of outliers is quite superficial. It's unclear why it's not always the case that the initial truncation does not always lead a 1 in the most significant bit: if the Max value of the block is used for quantization to 16-bit, it should be the case, however if a max on a much larger block is used, then it can very well be the case that the max will squeeze almost all the bits of an entire block to zero). Since the max will tend to be the outlier, then the loss of precision on all the numbers in a block could be massive.*
>
> **A1**: Please note that our method consists of two distinct stages: quantization and compression, with significantly different granularities. During the quantization stage, we utilize large size blocks, such as per-channel for weights, per-tensor for activations, and per-head for KV caches. In this stage, the largest outlier determines the most significant bit for the base precision scenarios (W8A16 or W8A16KV8).
>
> Following quantization, we apply compression within small size groups to address the concerns raised by the reviewer. We will further clarify it at the answer for Q1.

---

> ### Author Response · Authors · 2024-11-24
> **W2**
>
> **W2**: *There is no discussion on implementation on existing hardware The hardware idea proposed is discussed in isolation, leading to a inflation on the potential power advantages, since it neglects lots of HW that would not change (e.g. all the memory hieararchy). Comparison wrt SoA shows that improvements over fine-tuned previous work does not show super-impressive advantages.*
>
> **A2**:  It is true that many components in hardware are shared across operations, which means that the overall power savings may be less significant in practice. However, we emphasize that our contributions focus on improving the area and power efficiency of the MAC unit, which is one of the critical blocks in neural processing units.
>
> While direct comparisons of power consumption between QRazor and state-of-the-art (SOTA) methods like Quarot are challenging due to these shared hardware elements, we can indirectly demonstrate that QRazor offers significant computational benefits. Specifically, QRazor does not exhibit notable advantages over Quarot in terms of memory access. However, it provides a clear computational advantage.
>
> Our analysis of floating-point operations (FLOPs) and integer operations (OPs), presented in the Appendix **Figure 5** of our revised paper, shows that QRazor requires significantly fewer FLOPs—just 3% of those required by Quarot. Although QRazor incurs 35% more Ops than Quarot due to its compression technique, it is well-established that on GPUs, the power consumption of FLOPs (related to FP16 computations) is more than double that of OPs (related to INT8 computations). Consequently, this indicates that QRazor achieves substantially lower power consumption compared to Quarot, demonstrating its computational efficiency.

---

> ### Author Response · Authors · 2024-11-24
> **Q1**
>
> **Q1**: *It would be very good to see for each block which bit would be non-zero on average, i.e. on average how many leading bits can be truncated, also it would be good to see the percentage of zerooed elements per block after the aggressive quantization step.*
>
> **A3**: As you have proposed, we have experimented on possible number of truncation on average and non-zero value portion in each block. **Figure 2** in our revised paper demonstrates that the reviewer's concern is effectively addressed in our QRazor. It illustrates the leading '1' positions before 4-bit compression (immediately after converting to the sign-and-magnitude format). The leading '1' positions for activations are predominantly located between the 8th and 12th bit orders. For instance, if the leading '1' position in a group is the 13th bit, parameters with an MSB below the 9th bit will be zeroed after compression, as only 4 bits including sign bit are retained. Fortunately, the proportion of groups where the leading '1' position exceeds the 12th-bit order is minimal (only 9%), indicating that outliers are infrequent and mitigating concerns about significant parameters being truncated after 4-bit compression.
>
> We also analyzed the proportion of zeroed elements before and after 4-bit compression. For Q, K, and V, the increase in zeroed elements is not so considerable. However, for activations and weights, there is a notable increment. This is expected, as significant activations and weights often have small absolute values close to zero. Such truncation of small values does not lead to substantial errors. Combined with the low occurrence of outliers, as mentioned above, this explains why QRazor consistently delivers reliable performance across various LLMs, as discussed in Section 5 in our paper.

---

> ### Author Response · Authors · 2024-11-27
> **Official Comment by Authors**
>
> Dear Reviewer XPyT,
>
> Thank you for your review. If our response effectively resolves your issue, we'd be grateful if you consider improving your score.
>
> Thank you and wish you have a good end of the year!
>
> Authors

---

### Official Review · Reviewer_YqfQ · 2024-11-03

**Soundness:** 3
**Presentation:** 2
**Contribution:** 2
**Rating:** 6
**Confidence:** 2

**Summary:**

This paper proposes a 2 stage quantization to reduce the memory and compute requirements for LLMs. It quantizes the weights, activations and KV caches in an LLM in two stages. They first quantize the FP16 parameters to integers and then compress these integers using their SDR algorithm. They show the benefits of their method by comparing with other Quantization methods on Llama and Llama2 class of models on lm-eval-harness.

**Strengths:**

1.⁠ ⁠Their method is simple to understand and shows significant benefits to the baselines
2.⁠ ⁠They show empirically that QRazor performs much better than the baselines at the same bitwidth.

**Weaknesses:**

1.⁠ ⁠Since they keep the most significant bits from the razoring point, they are not effectively using the range of their bitwidth. This is because they use integer shifts instead of floating point scales.
2.⁠ ⁠Why are the FP16 accuracies for the same model different when comparing with baselines and QRazor in Table 1? This makes me question the fairness of the comparision.
3.⁠ ⁠Because of these, it will be good to include experiemental results with different tasks and models for comparision.
4.⁠ ⁠In table 1, what are the effective bit-widths of the baselines and QRazor?
5.⁠ ⁠How do the area and power numbers compare for baselines?

The paper presents a clear idea for quantizing LLMs, but can improve in the experimental benchmarking of their method and baselines.

**Questions:**

1.⁠ ⁠How does this quantization scheme affect the inference speed? Since the activations are quantized per token, I am assuming the inference speed takes a huge hit.
2.⁠ ⁠How does this method compare to weight-only-quantization methods like GPTQ, QuIP or FrameQuant? They show that activation quantization is not a huge bottleneck in LLM inference and are able to achive very low bitwidth for weights.

---

> ### Author Response · Authors · 2024-11-23
> **W1**
>
> Thank you for your review and questions regarding inference speed of our quantization scheme and the importance of activation quantization in LLM models! We would like to answer them one at a time. The revised parts of the paper are colored in blue. As we answer to your questions, we will cite the table or figure numbers which you can  look into for your better understanding. Please refer to our citations for more information of our answers.
>
> **W1**: *Since they keep the most significant bits from the razoring point, they are not effectively using the range of their bitwidth. This is because they use integer shifts instead of floating point scales.*
>
> **A1** : We consider that floating-point scale is similar to dynamic per-max quantization (DMQ). Although integer shifts are less efficient than DMQ in terms of bitwidth utilization, they enable significantly lighter compression and decompression. We further detailed the merit of our scheme compared to DMQ, as follows.
>
> 1. Eliminating Absolute Max Computation:
> Instead of determining the absolute maximum value within a group, QRazor detects only the leading '1'. While the absolute maximum inherently contains the leading '1', multiple parameters may share the same leading '1' position. In DMQ, identifying the absolute maximum is essential for computing the scaling factor. In contrast, our compression bypasses this step entirely, significantly reducing computational complexity.
>
> 2. Lightweight Compression and Decompression:
> Per-group DMQ typically involves group-level quantization and dequantization operations, which rely on arithmetic computations such as multiplication and, in some cases, division. In contrast, our compression and decompression rely on bit-level truncation and shifting, which are inherently simpler and far less resource-intensive.
>
> The above informations are clarified in our revised paper.

---

> ### Author Response · Authors · 2024-11-23
> **W2**
>
> **W2**: *Why are the FP16 accuracies for the same model different when comparing with baselines and QRazor in Table 1? This makes me question the fairness of the comparision.*
>
> **A2**: In the initial manuscript, we could not directly compare QRazor with Quarot due to transformers and datasets version compatibility issues. Specifically, we conducted our experiments using the opensource version of HuggingFace (Ejafa/llama_7B, NousResearch/Llama-2-7b-hf, etc) which was optimized for huggingface benchmark models, while Quarot used the original Meta-llama2 version. Consequently, we relied on indirect comparisons, which may have caused confusion to the reviewer.
>
> In this revision, we have fully updated the accuracy and perplexity results to use the same baseline, eliminating any confusion during comparisons and enabling direct evaluations against Quarot. The updated results, presented in Table 1 of the revised paper, demonstrate that QRazor outperforms Quarot (RTN) for both group sizes, g16 and g32, which correspond to the compression group sizes.

---

> ### Author Response · Authors · 2024-11-23
> **W3**
>
> **W3**: *Because of these, it will be good to include experiemental results with different tasks and models for comparision.*
>
> **A3**:  Thank you for the suggestions. As you have requested, we applied QRazor to Mistral-7B and Gemma2-2B, shown in **Table 2** of our revised paper, achieving solid accuracy results across these models. Experiment with Mistral-7B has also been carried out with other quantization method  such as "TesseraQ" which is one of the SOTA scheme. Comparison with methods is also available by **Table 10** in Appendix section of our revised paper and simplified Table.R1 below. These findings highlight QRazor’s ability to deliver reliable performance across diverse models and configurations.
>
> **Table.R1**
> | **Model**     | **#Bits**         | **Method**       | **PIQA** | **ARC-e** | **ARC-c** | **HellaSwag** | **Winogrande** | **Avg**  |
> |:---------------:|:-------------------|:------------------|:----------:|:-----------:|:-----------:|:---------------:|:----------------:|:----------:|
> |               |  FP16     | Baseline | 80.74| 81.56 | 50.16| 61.25 | 74.27  | 69.60  |
> |      **Mistral-7B**    |    W4A4   |  TesseraQ* | 72.19| 65.90 | 33.78 | 49.02 | 57.61 | 55.71 |
> |               |    W4A4(g16)   | QRazor| 77.75| 77.98| 45.48 | 57.70| 69.77|65.74|
> |               |    W4A4(g32)   |  QRazor| 77.80 | 75.39 | 43.47 | 56.20 | 68.90 | 64.35 |
>
> '*' means this version of TesseraQ is initialized from AWQ

---

> ### Author Response · Authors · 2024-11-23
> **W4**
>
> **W4**: *In table 1, what are the effective bit-widths of the baselines and QRazor?*
>
> **A4**: For our group size of 16 and 32 each has effective bit widhts of 4.25 and 4.125 which is calculated by function of
>
>  **{Number of quantized bits + (Scale Factor or Flag bit's bit width/group size)}**.
>
> During our compression stage, flag bit of 4(4bits) is created describing the number of truncated LSBs(Least Significant Bits) and number of shifts. This is equivalent to the case when floating point 16bit(4.125 effective bit) or 32bit(4.25 effective bit) is created when quantizing group size of 128, which is commonly used group size in other previous quantization works. As for the baseline, the effective bit width is 16 since it uses floating point 16bit data format.
>
> We applied QRazor to Mistral-7B and Gemma2-2B, shown in **Table 2** of our revised paper, achieving solid accuracy results across these models. These findings highlight QRazor’s ability to deliver reliable performance across diverse models and configurations.
>
> We have also attached **Table 4** which shows the accuracy of Llama models depending on the group size or effective bit width. Please refer to the **Tables** metioned for better understanding.

---

> ### Author Response · Authors · 2024-11-23
> **W5**
>
> **W5**: *How do the area and power numbers compare for baselines?*
>
> **A5**: Area and Power of baseline FP 16 x 16 MAC has been synthesized and added to the **Table 5** in the revised paper. Comared to 16 x 16 FP MAC, our proposed 4 x 4 INT MAC showed 85% area efficiency and 89% power saving. For convenient observation for the reviewr, Table.R2 is shown below.
>
> **Table.R2**
> |               | FP 16×16 MAC(Baseline)   | INT 16×8 MAC   | INT 8×8 MAC    | INT 4×4 Proposed |
> |---------------|:----------------:|:----------------:|:----------------:|:------------------:|
> | **Area (μm²)**|                |                |                |                  |
> | Multiplier    | 3042.2         | 1052.2         | 559.4          | 112              |
> | Additional Shifters | 0         | 0              | 0              | 156.5            |
> | Reg. + Accm.  | 1127.1         | 631            | 431            | 385.3            |
> | **Total**     | 4169.3         | 1683.2         | 990.4          | 653.8            |
> | **Power (mW)**|                |                |                |                  |
> | Multiplier    | 0.3378         | 0.0506         | 0.023          | 0.0028           |
> | Additional Shifters | 0         | 0              | 0              | 0.0067           |
> | Reg. + Accm.  | 0.1242         | 0.0733         | 0.0581         | 0.0451           |
> | **Total**     | 0.4620         | 0.1239         | 0.0811         | 0.0546           |

---

> ### Author Response · Authors · 2024-11-23
> **W6**
>
> **W6**: *The paper presents a clear idea for quantizing LLMs, but can improve in the experimental benchmarking of their method and baselines.*
>
> **A6**: We appreciate your constructive feedback. As you have said, we have fully revised all accuracy results and  standardized baselines for direct comparison with previous quantization methods. Even with the revised version, our quantization scheme shows reliable results.
>
> For more results of accuracy comparison with current SOTA works, please refer to **Table 1** and **Table 10** in our paper.

---

> ### Author Response · Authors · 2024-11-23
> **Q1**
>
> **Q1**: *How does this quantization scheme affect the inference speed? Since the activations are quantized per token, I am assuming the inference speed takes a huge hit.*
>
> **A7**:  At the quantization stage, we perform static per-tensor activation quantization (from FP16 to INT16) and static per-channel weight quantization (from FP16 to INT8), as clarified in our revised manuscript. After this stage, the quantized INT16 data is compressed to 4 bits based on the leading one position within groups, with group sizes of 16 or 32. While the per-tensor quantization follows conventional PTQ methods, the smaller group sizes used during the compression stage introduce a potential concern that inference speed might be affected. However, such fine-grained work can potentially be fused into the epilogue of the prior operations. This method ease the multiple compression process burden thereby improving the inference speed.
>
> In the context of dedicated hardware, QRazor avoids computationally intensive techniques such as reordering or rotations for data distribution. Instead, it relies on simple bit-level shifts and truncations, which are highly efficient to implement in hardware. We in such dedicated hardware scenario, QRazor can achieve much faster inference speeds compared to SOTA methods like Quarot, which requires orthogonal hadamard matrix operation causing computational overhead when not only quantization but also during dequantization process.
>
> To support this conjecture, we compared FLOPs and operations (Ops) between Quarot and QRazor. Our QRazor scheme incurs **35%** more OPs than Quarot, primarily due to the second-stage compression. However, QRazor requires only **3%** of the FLOPs used by Quarot throughout the attention layer. The comparison results are available in **Figure 5** of our revised paper. QRazor demonstrated significantly lower computational requirements, strongly reinforcing its potential efficiency in dedicated hardware implementations.

---

> ### Author Response · Authors · 2024-11-23
> **Q2**
>
> **Q2**: *How does this method compare to weight-only-quantization methods like GPTQ, QuIP or FrameQuant? They show that activation quantization is not a huge bottleneck in LLM inference and are able to achive very low bitwidth for weights*
>
> **A8**: Thank you for your question! To explain, it is true that for small batch sizes, weight access is the most significant bottleneck in LLM inference. However, recent studies have demonstrated that this bottleneck can be significantly alleviated by increasing the batch size. With larger batches, the memory access cost associated with weights is effectively divided across the batch, reducing its overall impact.
>
> In contrast, KV caches cannot be shared across batches and are continuously accumulated during inference. For this reason, recent works emphasize that quantizing KV caches is essential to address the memory wall problem in LLM inference. In light of this, we quantize KV caches to 4 bits in our approach.
>
> Activations, which scale proportionally with batch size, can also become problematic in scenarios with large batching. More critically, activations directly influence computation-bound performance. In weight-only quantization schemes, the quantized weights typically convert into floating-point formats because most processors lack support for integer-to-floating-point multiplications. As a result, the overall computation limit is determined by the data format of activations, which is FP16 in works like AWQ. Even with dedicated hardware support, such a multiplication is extreme and less efficient than integer multiplication.
>
> In large batch scenarios, the memory-bound bottleneck caused by weights is significantly mitigated, shifting the critical performance factor to computation speed. For this reason, we consider activation quantization to be crucial. Numerous studies[1][2][3] have thoroughly explained the importance and benefits of quantizing activations in LLM inference.
>
> [1] Xiao et al. SmoothQuant: Accuracte and Efficient Post-Training Quantization for Large Language Models. ArXiv, abs/2211.10438
>
> [2] Yuan et al. Reorder-based Post-training Quantization for Large Language Models. ArXiv abs/2304.01089
>
> [3] Zhao et al. Atom: Low-bit Quantization for Efficient and Accurate LLM Serving. ArXiv, abs/2310.19102

---

> ### Author Response · Authors · 2024-11-27
> **Official Comment by Authors**
>
> Dear Reviewer YqfQ,
>
> Thank you for your review. If our response effectively resolves your issue, we'd be grateful if you consider improving your score.
>
> Thank you and wish you have a good end of the year!
>
> Authors

---

### Official Review · Reviewer_xoaX · 2024-11-04

**Soundness:** 3
**Presentation:** 3
**Contribution:** 2
**Rating:** 5
**Confidence:** 4

**Summary:**

This paper presents a novel post-training quantization (PTQ) method, QRazor, designed for efficient low-bit quantization of large language models (LLMs). This technique aims to address the challenge of accuracy loss typically associated with aggressive quantization methods, especially at 4-bit precision. QRazor introduces a two-stage quantization and compression approach that first quantizes weights, activations, and KV caches to 8- or 16-bit integers using a method called absolute max scaling. It then applies a new compression technique, Significant Data Razoring (SDR), which selectively retains the most significant bits of data, reducing memory and computational demands without decompression. Experimental results show that QRazor achieves accuracy levels on par with state-of-the-art quantization methods.

**Strengths:**

Efficient Quantization: QRazor successfully enables 4-bit quantization for weights, activations, and KV caches while reducing the memory and computation costs of deploying LLMs in resource-constrained environments.

Adaptability Across Models: QRazor demonstrates effectiveness across different model types (e.g., LLaMA, LLaMA2) and quantization settings, showing flexibility and robustness in various use cases.

Extensive Evaluation: The paper provides detailed evaluations on task accuracy, resource usage, and power efficiency, supporting the method's applicability and benefits.

**Weaknesses:**

Accuracy Trade-offs: Although accuracy degradation is minimized, the paper does note some accuracy drops in specific tasks, especially under aggressive group compression settings. This could impact applications where even minor precision loss is critical.

Complexity of Scaling Parameters: The granularity of scaling parameters, such as per-channel and per-tensor configurations, introduces complexity in implementation and may require careful tuning to achieve optimal results.

**Questions:**

- How does the proposed method perform on fine-tuned LLMs for specific downstream tasks?

- It would be helpful to include the total number of parameters and FLOPs in all comparison tables for clearer evaluation of computational efficiency.

-It would be helpful to provide a direct comparison with prior works from the introduction section.

---

> ### Author Response · Authors · 2024-11-23
> **W1**
>
> Thank you for your review and recommendation! We would like to answer them one at a time. The revised parts of the paper are colored in blue. As we answer to your questions, we will cite the table or figure numbers for your better understanding. Please refer to our citations for more information of our answers. We have incorporated your suggestion into the revised paper, which dedicated to improvement of the paper's quality which we are sincerely grateful for.
>
> **W1**: *Accuracy Trade-offs: Although accuracy degradation is minimized, the paper does note some accuracy drops in specific tasks, especially under aggressive group compression settings. This could impact applications where even minor precision loss is critical.*
>
> **A1**:  Post-training quantization (PTQ) inevitably involves some degree of accuracy drop. However, it offers substantial power savings and hardware performance improvements, which is why it remains an active area of research. In our QRazor, we focus on group sizes of 16(g16) or 32(g32) while others(g64 and g128) are shown for the reference. Please, note that we use only 4-bit Flags for the compression. Many SOTA 4-bit LLM works use per-group scaling, where the group size is typically 128 and the scaling parameter is FP16 or FP32. For instance, Quarot quantized KV caches in such way. Considering this, g16 or g32 provides the same effective bit-width of 4.25 or 4.125 as the SOTA work which has group size of 128 and the scaling parameter is FP16 or FP32. In group size of 16 and 32, QRazor shows strong accuracy results compared to SOTA works.
>
> Although QRazor delivers competitive accuracies compared to state-of-the-art (SOTA) 4-bit LLM PTQ techniques, it is true that a certain degree of accuracy drop is inevitable when all parameters are quantized to 4-bit. This is a common limitation of approaches that fully quantize parameters to 4-bit precision. As a result, such an approach is not suitable for scenarios where even minor accuracy drops are unacceptable. However, this technique is highly applicable to use cases where moderate accuracy trade-offs are permissible, and resource constraints are stringent. QRazor effectively addresses such scenarios, making it a valuable contribution to the field.

---

> ### Author Response · Authors · 2024-11-23
> **W2**
>
> **W2**: *Complexity of Scaling Parameters: The granularity of scaling parameters, such as per-channel and per-tensor configurations, introduces complexity in implementation and may require careful tuning to achieve optimal results.*
>
> **A2**: Firstly, we sincerely apologize for not explicitly clarifying in our prior manuscript that our quantization stage—where FP16 models are quantized to **W8A16** or **W8A16KV8**—**employs only static scaling** for all parameters. This has been addressed in our revised paper. Static scaling is a widely adopted approach in PTQ techniques, as coarser granularity scaling parameters reduce computational complexity, as discussed in milestone works like SmoothQuant.
>
> Furthermore, we did not undertake significant tuning efforts in this paper, which still shows promising results. The details of our setup are provided in Section 5.  Calibration and validation of zero-shot accuracy and perplexity were conducted using 128 randomly selected samples from Wikitext2 across all tasks. Static quantization, utilizing absolute max scaling, was applied to weights, activations, and KV cache values. The scaling granularity was configured as per-channel for weights, per-tensor for activations, and per-head for KV cache values—this represents a typical PTQ setup and does not require extensive tuning efforts.
>
> For compression, we determine the leading one position for each group and then perform truncation and rounding. Weight compression is performed offline, while activations and KV caches are compressed online. This approach involves significantly less effort compared to other state-of-the-art (SOTA) methods. For example, GPTQ requires considerable fine-tuning for weight quantization, whereas our method eliminates such overhead.
>
> Given these considerations, we believe our methodology justifies the title of our paper: “QRAZOR: RELIABLE AND EFFORTLESS 4-BIT LLM QUANTIZATION BY SIGNIFICANT DATA RAZORING.

---

> ### Author Response · Authors · 2024-11-23
> **Q1**
>
> **Q1**: *How does the proposed method perform on fine-tuned LLMs for specific downstream tasks?*
>
> **A3**: We conducted experiments on two cases: BERT-large fine-tuned on Squad and Mistral-7B fine-tuned on Mental Health. The results, as shown in the following tables, demonstrate strong performance for the W4A4 and W4A4KV4 configuration across both cases.
>
> **Table.R4**
>
> | **Model**     | **#Bits**         | **Method**       | **SST2** | **RTE** | **QNLI** | **MRPC** | **MNLI** | **Avg**  |
> |:---------------:|:-------------------|:------------------|:----------:|:-----------:|:-----------:|:---------------:|:----------------:|:----------:|
> ||  FP16 | Baseline  | 49.08| 52.71| 49.46| 31.62| 35.45| 43.66|
> ||    W4A8(g16)  || 48.85| 52.48| 49.13| 31.78| 35.57| 43.56|
> |   **BERT-large-uncased-whole-word-masking-finetuned-squad** |    W4A8(g32) |QRazor| 48.51| 52.14| 49.09| 31.51| 35.46| 43.34|
> ||    W4A4(g16)   | | 48.11| 51.24| 49.35| 31.72| 34.45| 43.17|
> || W4A4(g32)   || 48.23| 51.32| 49.18| 31.67| 35.79| 43.24|
>
>
> **Table.R5**
>
> | **Model**     | **#Bits**         | **Method**       | **PIQA** | **ARC-e** | **ARC-c** | **HellaSwag** | **Winogrande** | **Avg**  |
> |:---------------:|:-------------------|:------------------|:----------:|:-----------:|:-----------:|:---------------:|:----------------:|:----------:|
> ||  FP16     | Baseline                 | 80.09| 81.40 | 44.15| 80.36 | 73.32| 71.86|
> ||    W4A8(g16)    |              | 80.63| 80.53| 43.81| 80.36| 73.01| 71.67|
> ||    W4A8(g32)    |              | 80.52| 80.35| 43.14| 80.46| 73.32| 71.56|
> ||    W4A8KV4(g16)   || 79.98| 80.00| 43.48| 79.81| 71.35| 70.92|
> |**Mistral-7B-Fintuned-Mental-Health**| W4A8KV4(g32)      |QRazor | 79.87| 79.30| 43.81| 79.90| 71.98| 70.97|
> ||    W4A4(g16)   |    | 79.49| 79.30| 43.14| 79.64| 71.59| 70.63|
> ||    W4A4(g32)   || 79.00| 77.89| 42.82| 79.25| 70.17| 69.83|
> ||    W4A4KV4(g16)   || 79.43| 80.18| 45.82| 78.84| 70.96| 71.05|
> || W4A4KV4(g32)      || 78.84| 78.77| 43.14| 78.43| 68.82| 69.60|

---

> ### Author Response · Authors · 2024-11-23
> **Q2**
>
> **Q2**: *It would be helpful to include the total number of parameters and FLOPs in all comparison tables for clearer evaluation of computational efficiency.*
>
> **A4**: Thank you again for the suggestion. Reviewer CcYw raised a similar question. Firstly, due to our two-stage quantization, one might assume that the total number of parameters is effectively doubled, leading to increased memory requirements. However, it is important to clarify that the interim parameters generated during the first-stage quantization are only used once. By performing the second-stage compression immediately after the first-stage quantization, this concern is effectively addressed. Please, refer to the answer for the reviewer 1’s question.
>
> To further evaluate the computational overhead, we quantified the number of floating-point operations (FLOPs) and integer operations (OPs). For reference, we conducted the same analysis for Quarot, the state-of-the-art (SOTA) 4-bit LLM. The results, presented in **Figure 5** of Appendix, compare the computational efficiencies of Quarot and QRazor.
>
> QRazor incurs **35%** more OPs than Quarot, primarily due to the second-stage compression. However, QRazor requires only **3%** of the FLOPs used by Quarot. This significant difference arises because Quarot relies on Hadamard matrix multiplication to rotate activations and KV caches, which inflates its FLOP count. Considering that FLOPs are computationally heavier than OPs, we conclude that QRazor demonstrates substantially better computational efficiency compared to Quarot.
>
> We compare computational efficiency exclusively with Quarot, as it is the only method that demonstrates accuracy comparable to QRazor.

---

> ### Author Response · Authors · 2024-11-23
> **Q3**
>
> **Q3**: *It would be helpful to provide a direct comparison with prior works from the introduction section.*
>
> **A5**: We added the following sentences to the introduction as requested. “The accuracies of these configurations are compared against state-of-the-art (SOTA) 4-bit LLMs. For the LLaMA-1-7B and LLaMA-1-13B models, QRazor demonstrates superior performance, achieving more than a 10% improvement in accuracy compared to QLLM. When compared to Quarot, another SOTA method, QRazor achieves significantly better results when evaluated against the rounding-to-nearest (RTN) baseline, and nearly matching results for configurations involving GPTQ for the LLaMA-2-7B and -13B models.”
>
> We hope this introduction could provide forehand  information to the reviewers for direct comparison with prior SOTA works and clarify the contribution of our work.

---

> ### Author Response · Authors · 2024-11-27
> **Official Comment by Authors**
>
> Dear Reviewer xoaX,
>
> For your queries, we carefully prepared the answers. We look forward to you considering raising your score if our response effectively addressed your question.
>
> Thank you and hope you have a good end of the year!
>
> Authors

---

### Official Review · Reviewer_CcYw · 2024-11-05

**Soundness:** 3
**Presentation:** 4
**Contribution:** 3
**Rating:** 6
**Confidence:** 2

**Summary:**

QRazor presents a two-stage quantization process (quantization and compression) that allows for reliable 4-bit quantization of weights, activations, and KV cache in transformer-based LLMs

**Strengths:**

+ This technique retains only the four most salient bits while discarding others, enabling effective compression without manipulating data distributions

+ QRazor achieves LLM accuracies that are better or comparable to state-of-the-art 4-bit methods, despite reduced quantization effort

+ The paper presents an integer-based arithmetic unit specifically designed for QRazor, allowing for direct low-precision arithmetic operations without decompressing the SDR data

**Weaknesses:**

While the paper claims superior or comparable performance to state-of-the-art methods, it would be beneficial to see a more comprehensive comparison with other recent quantization techniques. I know there is a lot of quantization papers on LLMs and it's hard to keep up with it. But, it ll be good to at least compare with some recent works that are showing SOTA results: QuaRot [1], TesseraQ [2].

How will Qrazor perform on architectures beside Llama, say mistral or gemma? What about llama 3.2 specifically the edge versions: 1B, 3B models?

While the paper discusses hardware efficiency, it doesn't provide a detailed analysis of any potential computational overhead introduced by the two-stage quantization process. Also, is there an impact from a memory perspective?

[1] Ashkboos, Saleh, et al. "Quarot: Outlier-free 4-bit inference in rotated llms." arXiv preprint arXiv:2404.00456 (2024).

[2] Li, Yuhang, and Priyadarshini Panda. "TesseraQ: Ultra Low-Bit LLM Post-Training Quantization with Block Reconstruction." arXiv preprint arXiv:2410.19103 (2024).

**Questions:**

See weaknesses above

---

> ### Author Response · Authors · 2024-11-23
> **W1**
>
> Thank you for your review and recommendation! We would like to answer them one at a time. The revised parts of the paper are colored in blue. As we answer to your questions, we will cite the table or figure numbers for your better understanding. Please refer to our citations for more information of our answers.
>
> **Weakness**
>
> **W1**: *While the paper claims superior or comparable performance to state-of-the-art methods, it would be beneficial to see a more comprehensive comparison with other recent quantization techniques. I know there is a lot of quantization papers on LLMs and it's hard to keep up with it. But, it ll be good to at least compare with some recent works that are showing SOTA results: QuaRot [1], TesseraQ [2].*
>
> **A1**: In the initial manuscript, we could not directly compare QRazor with Quarot due to transformers and datasets version compatibility issues. Specifically, we conducted our experiments using the opensource version of HuggingFace which was optimized for huggingface benchmark models, while Quarot used the original Meta-llama2 version. Consequently, we relied on indirect comparisons, which may have caused confusion among reviewers.
>
> In this revision, we have fully updated the accuracy and perplexity results to use the same baseline, eliminating any confusion during comparisons and enabling direct evaluations against Quarot. The updated results, presented in  **Table 1** of the revised paper and Table.R1 below, demonstrate that QRazor outperforms Quarot (RTN) for both group sizes, g16 and g32, which correspond to the compression group sizes. When compared to Quarot (GPTQ), QRazor achieves slightly lower accuracies.
>
> However, the computational efficiency of QRazor is significantly better than that of Quarot, as detailed in the response for W3. Therefore, we consider QRazor to be a more efficient technique than Quarot (GPTQ), despite its slightly lower accuracy. Furthermore, when compared to Quarot (RTN), QRazor outperforms in both accuracy and computational efficiency.
>
> **Table.R1**
>
> | **Model**     | **#Bits**         | **Method**       | **PIQA** | **ARC-e** | **ARC-c** | **HellaSwag** | **Winogrande** | **Avg**  |
> |:---------------:|:-------------------|:------------------|:----------:|:-----------:|:-----------:|:---------------:|:----------------:|:----------:|
> |               |  FP16     | Baseline | 79.13| 74.39 | 45.97| 76.21 | 69.30  | 69.00  |
> |               |    W4A4KV4    | QuaRot(RTN)| 72.09| 58.88 | 35.24| 65.40 | 60.69 | 58.26|
> |      **Llama-2-7B**    |    W4A4KV4   |  QuaRot(GPTQ) | 76.77| 69.87 | 40.87 | 72.16 | 63.77 | 65.64 |
> |               |    W4A4KV4(g16)   | QRazor| 73.39| 70.88| 39.80 | 70.15| 64.01|63.65|
> |               |    W4A4KV4(g32)   |  QRazor| 73.23 | 67.54 | 37.46 | 67.16 | 60.46 | 61.17 |
> |---------------|-------------------|------------------|----------|-----------|-----------|---------------|----------------|----------|
> |               |  FP16     | Baseline                 | 80.23 | 77.82 | 48.76| 79.39 | 72.30| 71.70|
> |               |    W4A4KV4     | QuaRot(RTN)| 77.37 | 70.83| 43.69| 73.11| 67.32| 67.16 |
> |     **Llama-2-13B**   |W4A4KV4| QuaRot(GPTQ)| 78.89| 72,98| 46.59| 76.37| 70.24| 69.79 |
> |               |    W4A4KV4(g16)   |QRazor| 77.80| 75.44| 44.15| 76.06 | 67.72| 68.23 |
> |               |    W4A4KV4(g32)   |QRazor| 77.09| 74.91| 44.28| 75.06 | 66.77| 67.62 |
>
>
> We also note that TerresaQ is a recent work that we overlooked in the initial submission. Upon review, TerresaQ appears to be a solid contribution. In Appendix of our revised paper, we include a detailed comparison between QRazor and TerresaQ with Mistral-7B model in Appendix **Table 10**. For the W4A4 configuration, QRazor outperforms TerresaQ as it is shown in **Table.R2** below.
>
> These direct comparisons indicate that QRazor achieves robust performance relative to state-of-the-art techniques.
>
> **Table.R2**
> | **Model**     | **#Bits**         | **Method**       | **PIQA** | **ARC-e** | **ARC-c** | **HellaSwag** | **Winogrande** | **Avg**  |
> |:---------------:|:-------------------|:------------------|:----------:|:-----------:|:-----------:|:---------------:|:----------------:|:----------:|
> |               |  FP16     | Baseline | 80.74| 81.56 | 50.16| 61.25 | 74.27  | 69.60  |
> |      **Mistral-7B**    |    W4A4   |  TesseraQ* | 72.19| 65.90 | 33.78 | 49.02 | 57.61 | 55.71 |
> |               |    W4A4(g16)   | QRazor| 77.75| 77.98| 45.48 | 57.70| 69.77|65.74|
> |               |    W4A4(g32)   |  QRazor| 77.80 | 75.39 | 43.47 | 56.20 | 68.90 | 64.35 |
>
> '*' means this version of TesseraQ is initialized from AWQ

---

> ### Author Response · Authors · 2024-11-23
> **W2**
>
> **W2**: *How will Qrazor perform on architectures beside Llama, say mistral or gemma? What about llama 3.2 specifically the edge versions: 1B, 3B models?*
>
> **A2**: We applied QRazor to Mistral-7B, achieving solid accuracy results across these models, as shown in **Table.R3** below and **Table 2** of our revised paper.
> For Gemma2-2B, which have not yet been addressed by prior 4-bit LLM works to our knowledge, QRazor demonstrates very strong accuracy results. These findings highlight QRazor’s ability to deliver reliable performance across diverse models and configurations.
>
> **Table.R3**
>
> | **Model**     | **#Bits**         | **Method**       | **PIQA** | **ARC-e** | **ARC-c** | **HellaSwag** | **Winogrande** | **Avg**  |
> |:---------------:|:-------------------:|:------------------:|:----------:|:-----------:|:-----------:|:---------------:|:----------------:|:----------:|
> |               |  FP16     | Baseline | 60.17| 37.89 | 22.41| 35.09 | 51.07  | 41.32  |
> |               |    W4A4(g16)    |          | 57.02    | 38.77     | 23.08     | 32.82         | 51.78          | 40.96    |
> |      **Gemma-2-2B**    |    W4A4(g32)   |  QRazor              | 55.71    | 32.46     | 20.06     | 32.13         | 51.05          | 38.28    |
> |               |    W4A4KV4(g16)   |               | 57.18    | 37.54     | 19.73     | 32.80         | 50.51          | 39.55    |
> |               |    W4A4KV4(g32)   |               | 55.98    | 33.16     | 20.74     | 32.15         | 47.91          | 37.99    |
> |---------------|-------------------|------------------|----------|-----------|-----------|---------------|----------------|----------|
> |               |  FP16     | Baseline                 | 80.74    | 81.58     | 50.16     | 61.25        | 74.27         | 69.60    |
> |               |    W4A4(g16)    |                     | 77.75    | 77.98     | 45.48     | 57.70         | 69.77          | 65.74    |
> |     **Mistral-7B**   |    W4A4(g32)   | QRazor              | 77.80    | 75.39     | 43.47     | 56.20         | 68.90          | 64.35    |
> |               |    W4A4KV4(g16)   |               | 77.78    | 77.34     | 44.48     | 58.09         | 68.56          | 65.25    |
> |               |    W4A4KV4(g32)   |               | 77.53    | 76.09    | 42.24     | 56.38         | 66.38          | 63.72    |
>
> However, for Llama 3.2-1B and 3B models, QRazor exhibits noticeable accuracy drops for W4A4 with group size = 16 as it is shown on **Table.R4**. By reducing the group size to 8, we observed moderate accuracy improvements for these models. While QRazor experiences some challenges with these specific edge models, it demonstrates very solid accuracy on another edge model, Gemma2-2B. Further, for W4A8, QRazor shows solid accuracies. This suggests potential for further refinement to improve performance on Llama 3.2-1B and 3B, which we identify as an area for future work. The quantization of KV caches has not been attempted since accuracy drops are observed at the scenario for W4A4.
>
> **Table.R4**
> | **Model**     | **#Bits**         | **Method**       | **PIQA** | **ARC-e** | **ARC-c** | **HellaSwag** | **Winogrande** | **Avg**  |
> |:---------------:|:-------------------:|:------------------:|:----------:|:-----------:|:-----------:|:---------------:|:----------------:|:----------:|
> |               |  FP16     | Baseline                 | 74.65    | 64.21     | 31.77     | 63.73         | 59.83         | 58.84    |
> |               |    W4A8(g8)    |              | 72.25    | 61.70     | 31.10     | 60.21         | 59.01          | 56.85   |
> |               |    W4A8(g16)    |              | 70.91    | 60.12     | 29.76     | 59.21         | 58.59          | 55.72    |
> |    **Llama-3.2-1B**  |    W4A8(g32)   |   QRazor  | 71.23    | 59.97     | 28.09     | 58.18         | 55.12          | 54.52    |
> |               |    W4A4(g8)   |               | 68.52   | 57.84     | 25.75     | 53.23         | 56.06          | 52.28    |
> |               |    W4A4(g16)   |               | 62.25    | 49.07    | 23.74     | 46.29         | 55.56          | 47.38    |
> |               | W4A4(g32)      |               | 57.95   | 37.67  | 22.06 | 37.75 | 52.62    | 41.61   |
> |---------------|-------------------|------------------|----------|-----------|-----------|---------------|----------------|----------|
> |               |  FP16     | Baseline  | 76.61    | 74.91 | 43.14  | 73.49 | 69.53 | 67.53   |
> |               |    W4A8(g8)  |             | 75.52| 71.93 | 36.45| 71.49| 67.09| 64.50|
> |               |    W4A8(g16) |             | 75.30| 70.53| 36.45 | 70.43 | 66.30 | 63.80 |
> |    **Llama-3.2-3B**  |    W4A8(g32)   | QRazor| 75.41| 68.89| 35.45| 69.29| 66.69| 63.15 |
> |               | W4A4(g8)   |              | 74.01| 63.86| 33.10| 66.24| 64.52| 60.35|
> |               | W4A4(g16) |               | 72.12| 63.09| 31.71| 62.23| 62.85| 58.40|
> |               | W4A4(g32) |              | 69.39| 57.18| 28.97| 57.94| 57.04| 54.11|

---

> ### Author Response · Authors · 2024-11-23
> **W3**
>
> **W3**: *While the paper discusses hardware efficiency, it doesn't provide a detailed analysis of any potential computational overhead introduced by the two-stage quantization process. Also, is there an impact from a memory perspective?*
>
> **A3**: For the second-stage compression, we convert the 2’s complement format of the base precision (W8A16 or W8A16KV8) to a sign-and-magnitude representation. Then, we detect the leading one position for each group using bitwise OR operations, followed by truncation and rounding. We quantified the computational overhead by counting the number of floating-point operations (FLOPs) and integer operations (OPs). For reference, we conducted the same analysis for Quarot. Based on these results, we compared the computational efficiencies of Quarot and QRazor, as shown in Appendix **Figure 5** of our revised paper.
>
> QRazor incurs **35%** more OPs compared to Quarot, primarily due to the second-stage compression. However, QRazor requires only **3%** of the FLOPs needed by Quarot. This stark contrast arises because Quarot relies on Hadamard matrix multiplication to rotate activations and KV caches, which significantly inflates its FLOP count. These results highlight QRazor’s computational efficiency, even with the two-stage quantization process.
>
> The effective bit-width of QRazor is slightly larger than 4 bits, as shown in **Table 1**. Consequently, its impact on memory access is expected to be insignificant. Due to our two-stage quantization, one might be concerned that the memory requirement could be higher. However, it is important to note that the interim parameters generated during the first-stage quantization are only used once. By performing the second-stage compression immediately after the first-stage quantization, we can avoid that the interim parameters are stored to external DRAM. So, the above concern can be effectively mitigated.

---

> ### Author Response · Authors · 2024-11-27
> **Official Comment by Authors**
>
> Dear Reviewer CcYw,
>
> Thank you for your review. If our response effectively resolves your issue, we'd be grateful if you consider improving your score.
>
> Thank you and wish you have a good end of the year!
>
> Authors

---

### Author Response · Authors · 2024-11-25
**To Reviewers and Chairs**

Hello Reviewers and Chairs,

We sincerely appreciate your invaluable comments, which have significantly enhanced our work. As highlighted in our title, “QRazor: Reliable and Effortless 4-bit LLM Quantization by Significant Data Razoring,” our primary goal is to develop an efficient and straightforward 4-bit quantization method that achieves robust accuracy while remaining accessible for practical use.

Recent advancements in aggressive LLM quantization research have shown great promise in addressing the memory and energy bottlenecks inherent in LLMs. While aggressive quantization mitigates these issues, many existing methods introduce substantial computational overhead, negating their energy efficiency. Furthermore, the complexity of these techniques often necessitates expert implementation, limiting their practicality.

To tackle these challenges, QRazor is designed as a simple yet effective solution. Our approach builds on conventional PTQ schemes, employing static per-channel scaling for weights, static per-tensor scaling for activations, and static per-head scaling for KV caches. This ensures that floating-point parameters are initially quantized to high-precision integers without sacrificing accuracy. In the second stage, we apply lossy compression to further reduce all parameters to 4 bits. Despite requiring dynamic adjustments during compression, our method minimizes overhead by utilizing lightweight operations such as bitwise OR and logical shifts.

One reviewer raised a valid concern regarding the similarity between QRazor's method and the power-of-two scaling concept. However, the novelty of QRazor lies not in its final data format but in its ability to efficiently derive low-precision representations from high-precision integers while preserving a wide dynamic range. This approach fundamentally shifts the paradigm, challenging the traditional notion that quantization solely converts floating-point data types to integers. By focusing on simplicity, speed, and low overhead, QRazor ensures practical implementation without sacrificing performance.

Our experimental results clearly demonstrate that QRazor achieves strong accuracy across various LLMs, addressing the practical challenges of complexity and computational cost. QRazor provides a reliable, easy-to-implement quantization method suitable for resource-constrained scenarios, offering significant contributions to the field.

Once again, we thank the reviewers for their insightful feedback, which has been instrumental in refining our work.

Best Regards,
QRazor Research Team

---

### Meta-Review · Area_Chair_Fz6p · 2024-12-21

**Metareview:**

This paper proposed QRazor a post-training quantization (PTQ) method for 4-bit quantization of LLMs. The key idea of QRazor is a two-stage process combining quantization and compression. It introduced “Significant Data Razoring” (SDR) to preserve the most significant bits while discarding others.

Strengths:

1. The idea of QRazor using quantization and compression is interesting.

2. The method integrates with hardware-optimized integer arithmetic units, which has area and power savings.

3. Experimental results show that the proposed QRazor performs competitively on multiple benchmarks and architectures.

Weaknesses:

1.  The accuracy improvements over SOTA methods were marginal and did not justify the increased complexity of SDR.


2.  The statement on hardware efficiency ignores some broader system-level factors  such as memory hierarchy, with a special focus on  the MAC unit.

3.  Some reviewers concerned that truncating lower bits based on the most significant bit could lead to precision loss, especially for larger blocks.

4.  The experiments are insufficient. Concerns remain on the baselines used in this paper.

While QRazor is an interesting approach,I have to recommend rejection based on the weaknesses raised above. However, I sincerely encourage the authors to revise if before future submission.

**Additional Comments On Reviewer Discussion:**

The authors addressed some concerns while some still remain, e.g., potential precision loss due to the truncation method in SDR, insufficient analysis, inconsistent comparisons.

---

### Decision · Program_Chairs · 2025-01-22

Reject